# Intratumoral immunotherapy using platelet-cloaked nanoparticles enhances antitumor immunity in solid tumors

Baharak Bahmani[1,4], Hua Gong [2,4], Brian T. Luk [1,4], Kristofer J. Haushalter[1], Ethel DeTeresa[1], Mark Previti[1], Jiarong Zhou[2], Weiwei Gao [2], Jack D. Bui[3], Liangfang Zhang [2], Ronnie H. Fang [2✉] & Jie Zhang [1✉]

Intratumoral immunotherapy is an emerging modality for the treatment of solid tumors. Toll-like receptor (TLR) agonists have shown promise for eliciting immune responses, but systemic administration often results in the development of adverse side effects. Herein, we investigate whether localized delivery of the TLR agonist, resiquimod (R848), via platelet membrane-coated nanoparticles (PNP-R848) elicits antitumor responses. The membrane coating provides a means of enhancing interactions with the tumor microenvironment, thereby maximizing the activity of R848. Intratumoral administration of PNP-R848 strongly enhances local immune activation and leads to complete tumor regression in a colorectal tumor model, while providing protection against repeated tumor re-challenges. Moreover, treatment of an aggressive breast cancer model with intratumoral PNP-R848 delays tumor growth and inhibits lung metastasis. Our findings highlight the promise of locally delivering immunostimulatory payloads using biomimetic nanocarriers, which possess advantages such as enhanced biocompatibility and natural targeting affinities.

[1] Cello Therapeutics, Inc., San Diego, CA 92121, USA. [2] Department of NanoEngineering, University of California San Diego, La Jolla, CA 92093, USA. [3] Department of Pathology, University of California San Diego, La Jolla, CA 92093, USA. [4] These authors contributed equally: Baharak Bahmani, Hua Gong, Brian T. Luk. ✉email: rhfang@ucsd.edu; jie.zhang@cellothera.com

mmunotherapy has emerged as an effective therapeutic approach against cancer that harnesses the power of immune cells in the tumor microenvironment. Some recent approaches, including the use of immune checkpoint inhibitors against cytotoxic T-lymphocyte-associated protein 4 and programmed cell death protein 1 (Refs. [1,2]), as well as the adoptive transfer of chimeric antigen receptor (CAR) T cells[3], have shown considerable promise. Despite the clinical success of such immunotherapies in the treatment of various cancer types[4–7], each one still has its disadvantages that need to be overcome. For example, CAR T cell therapy has performed well against certain hematological cancers, but does not fare well against solid tumors[8]. Checkpoint blockade therapy is oftentimes associated with severe systemic side effects and only benefits a subset of patients with tumors that are in the correct immunological state[9,10]. One promising strategy to further expand the field of immunotherapy is the modulation of the tumor microenvironment via engagement of Toll-like receptors (TLRs) and inhibiting tumor-promoting immune signaling[11–14]. TLRs are mainly expressed by immune cells, and among them TLR7, an endosomal single-stranded RNA receptor, is predominantly expressed by macrophages, plasmacytoid dendritic cells, natural killer cells, and B cells[15].

Resiquimod (R848), a small-molecule immunomodulator, belongs to the TLR7/8 agonist family. Upon binding of R848 to TLR7/8, multiple immunomodulatory cytokines, including interleukin 6 (IL-6), IL-12, and interferon α (IFNα) are released, therefore triggering a cascade of signaling pathways that leads to the activation of antigen-presenting cells (APCs) and polarization of T cell responses[16–18]. Despite extensive investigations into the role of TLRs in inducing innate immune responses to bacterial and viral pathogens, only recently has attention shifted to their role in anticancer immunosurveillance. TLR7/8 signaling can promote anticancer responses through activation of the central transcription factor nuclear factor κB (NF-κB)[19]. It has been reported that TLR7/8 therapy leads to the expansion of tumor antigen-specific CD8$^+$ T cells, which is important for the development of an effective antitumor immune response[20,21].

Although the systemic administration of R848 and other members of the TLR7 agonist family in combination with checkpoint inhibitors has proven advantageous in the treatment of squamous cell carcinoma, colon carcinoma, metastatic melanoma, and pancreatic cancer[18,22–24], there are drawbacks limiting their clinical translation. For example, safety concerns were raised when multiple intravenous doses or oral administrations of small-molecule TLR7 agonists caused adverse events such as fever, fatigue, headache, and hypertension in patients[25–28]. In addition, some reports suggested that systemic administration of R848 leads to rapid depletion of leukocytes and transient local immune insufficiency[29]. To overcome these challenges, intratumoral injection of TLR7 agonists has been investigated as a more clinically relevant route of administration to address solid tumors[30–34]. The localization of immunostimulatory agents to the tumor microenvironment can convert it from a "cold" to a "hot" state, helping to kickstart antitumor immunity[35]. In order for intratumoral immunotherapies to be effective, it is necessary to confine the immune agonist payloads within the tumor site. However, the direct injection of free drug has the potential for systemic leakage that can lead to reduced efficacy, whereas targeted nanodelivery platforms are generally designed to be antigen-specific[36], limiting their broad applicability.

Herein we report on the development of a platelet membrane-cloaked nanoparticle (PNP) specifically for the intratumoral delivery of R848 to treat solid tumors. The plasma membrane derived from human platelets, with its multitude of proteins, glycoproteins, and lipids, bestows platelet-mimicking properties such as selective adherence to cells in the tumor microenvironment[37]. Cell membrane coating is a facile approach for improving biocompatibility while enabling nanoparticle platforms to effectively interface with biological targets, such as tumors, through multimodal interactions[38]. We demonstrate that R848-loaded PNP (PNP-R848) exhibits prolonged retention at the tumor site and improves cellular interactions within the tumor microenvironment. This enables the nanoformulation to exert significant biological activity upon intratumoral administration, even at low R848 dosages that would otherwise be ineffective when administered systemically. In an MC38 murine colorectal adenocarcinoma model, it is shown that PNP-R848 promotes the strong activation of APCs within the draining lymph node (DLN) and increases immune infiltration. This ultimately leads to a potent antitumor response that facilitates the complete short-term rejection of established tumors while bestowing long-term immunity that protects against repeated and highly aggressive tumor re-challenges. The antitumor activity of the formulation is further confirmed in a metastatic 4T1 murine triple-negative breast cancer model.

## Results

**Nanoparticle synthesis and characterization.** Given the multitude of interactions of platelets with other cell types and tissues[39–44], we aimed to leverage these unique abilities to design a nanoparticle platform incorporating natural targeting abilities. This was done by directly coating the membrane isolated from human platelets through a differential centrifugation and freeze-thaw process onto biocompatible and biodegradable polylactic acid (PLA) nanoparticle cores via sonication[37], an energetically favorable process that promotes colloidal stabilization[45]. The presence of phosphatidylserine, P-selectin, GPIbα, and integrin αIIbβ3 was confirmed on the surface of the platelet membrane ghosts by flow cytometry (Fig. 1a). Phosphatidylserine, P-selectin, and the full αIIbβ3 complex are expressed on the membrane surface upon platelet activation[46–49]. GPIbα is responsible for von Willebrand factor-mediated platelet adhesion[50]. P-selectin, αIIbβ3, and GPIbα have all been implicated in cancer pathogenesis, suggesting important interactions with tumor cells[51]. Despite the activation state of the membrane, additional assays for thrombin and adenosine diphosphate (ADP) verified the successful physical removal of these intracellular platelet-activating molecules responsible for propagating thrombotic responses, thus mitigating safety concerns (Fig. 1b, c). Another detailed characterization of the platelet membrane has been previously reported[37]. Physicochemical characterizations revealed that membrane coating slightly increased the hydrodynamic size of both the bare PLA nanoparticle cores as well as the bare R848-loaded nanoparticle cores (NP-R848) (Fig. 1d). Additionally, the surface zeta potential was similar between all samples (Fig. 1e). Transmission electron microscopy revealed that, compared to bare NP-R848, the final PNP-R848 formulation possessed a core–shell structure with a layer of membrane coating on the outside clearly separated from the nanoparticle core (Fig. 1f, g and Supplementary Fig. 1a). The nanoparticles remained stable in phosphate-buffered saline (PBS) over the course of 4 weeks, both when stored at room temperature and at 4 °C (Supplementary Fig. 1b). Quantification of drug loading revealed that 3.4 wt% of R848 could be encapsulated into the final formulation. Finally, the release of the R848 payload was studied over time, and the profiles for both the bare NP-R848 and coated PNP-R848 formulations matched closely, where more than 60% of the encapsulated payload was released within the first 24 h (Fig. 1h).

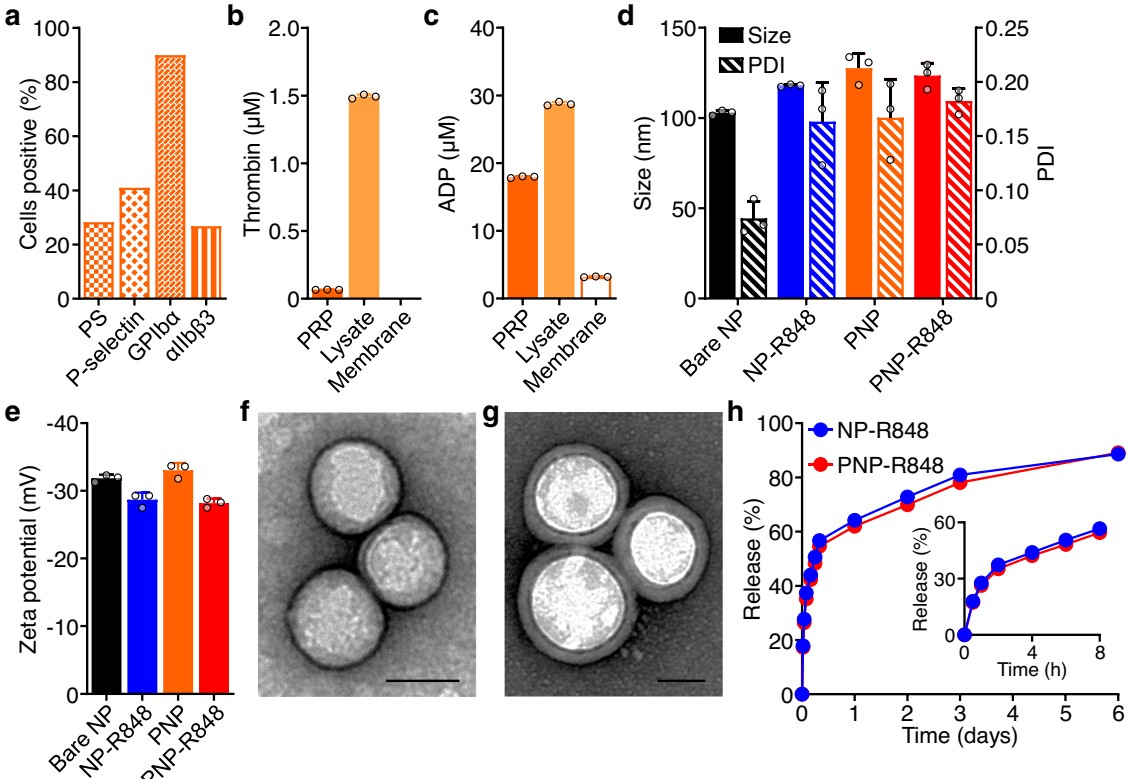

**Fig. 1 Nanoparticle characterization. a** Characterization of surface markers on platelet membrane, including phosphatidylserine (PS), P-selectin, GPIbα, and αIIbβ3. **b, c** Quantification of pro-thrombotic platelet-activating molecules thrombin (**b**) and adenosine diphosphate (ADP, **c**) in platelet-rich plasma (PRP), platelet lysate, and purified platelet membrane ($n = 3$, mean + SD). **d** Average hydrodynamic diameter and polydispersity index (PDI) of bare nanoparticle (NP) cores, uncoated NP-R848, PNP, and PNP-R848 ($n = 3$, mean + SD). **e** Zeta potential of bare NP, NP-R848, PNP, and PNP-R848 ($n = 3$, mean + SD). **f, g** Transmission electron microscopy visualization of uncoated NP-R848 (**f**) and coated PNP-R848 (**g**) with uranyl acetate negative staining (scale bars = 50 nm; repeated 3 times). **h** Drug release profile from uncoated NP-R848 and coated PNP-R848 over 6 days (3 independent experiments). Source data are provided as a Source Data file.

**Nanoparticle interaction with tumors**. In order to assess the interaction of PNP with solid tumor cell types, we studied both binding and uptake in vitro. Fluorescent dye-labeled nanoparticles were incubated with a panel of murine and human cancer cells, including MC38, HT-29, 4T1, and MDA-MB-231, at 4 °C for the binding study and 37 °C for the uptake study. It was observed by flow cytometry that PNP much more readily bound to all four cancer cells compared with a polyethylene glycol (PEG)-coated nanoparticle (PEG-NP) control (Fig. 2a). These results correlated well with cellular uptake, which was also significantly higher for PNP than for PEG-NP in all of the cell lines (Fig. 2b). Considering the enhanced interaction of PNP with MC38 cells in vitro, we next tested the retention time of PNP in an MC38 tumor model in vivo. After allowing the tumors to establish, mice received a single intratumoral administration of dye-labeled PEG-NP or PNP, and the nanoparticles were tracked using a live imaging system over the course of 7 days (Fig. 2c, d). Initially, there was a similar drop in the amount of nanoparticles present within the tumor. As time progressed, the difference between the two groups increased, and the greatest contrast was observed at 48 h, where on average 35% of the PNP remained, while only 11% of the PEG-NP was retained within the tumor. We then analyzed in vivo drug levels after intratumoral administration of R848-loaded nanoparticles. Compared with injection of free R848, which resulted in a large transient spike in serum R848 concentration, both PEG-NP-R848 and PNP-R848 were able to significantly limit systemic leakage (Fig. 2e). However, only PNP-R848, with its enhanced affinity to tumor cells, enabled prolonged drug persistence in the tumor tissue (Fig. 2f). Taken

together, these studies demonstrated that the platelet membrane coating, which displays surface markers known to play a role in cancer cell binding[52], was able to significantly increase nanoparticle affinity to the MC38 tumor cells compared with a more traditional PEG coating.

**In vitro immunostimulatory activity**. To directly assess the biological activity of the R848 payload, we incubated PNP-R848 with human reporter cell lines expressing either TLR7 or TLR8, which provide a colorimetric readout in response to NF-κB activation (Fig. 3a, b). The cells were incubated with free R848 or PNP-R848 for 21 h, and our results showed that the activities of the two were roughly equivalent at the same drug concentration. As expected, PNP nanoparticles without drug loading showed minimal TLR7 and TLR8 activation. We next studied the biological effect of PNP-R848 on bone marrow-derived cells (BMDCs), and it was observed that the formulation could induce the upregulation of CD80 and CD86, two APC maturation markers that serve as co-stimulatory signals for mediating downstream immune responses (Fig. 3c, d). The expression levels of CD80 and CD86 were comparable to those induced by free R848, indicating that the loading of the payload into the nanoparticles did not affect their potent immunomodulatory activity. Additionally, we assessed the ability of PNP-R848 to elicit the production of proinflammatory cytokines such as IL-6, tumor necrosis factor α (TNFα), and IL-12 by BMDCs (Fig. 3e–g). After incubation with various concentrations of free R848 or PNP-R848, the culture supernatant was analyzed by enzyme-linked immunosorbent

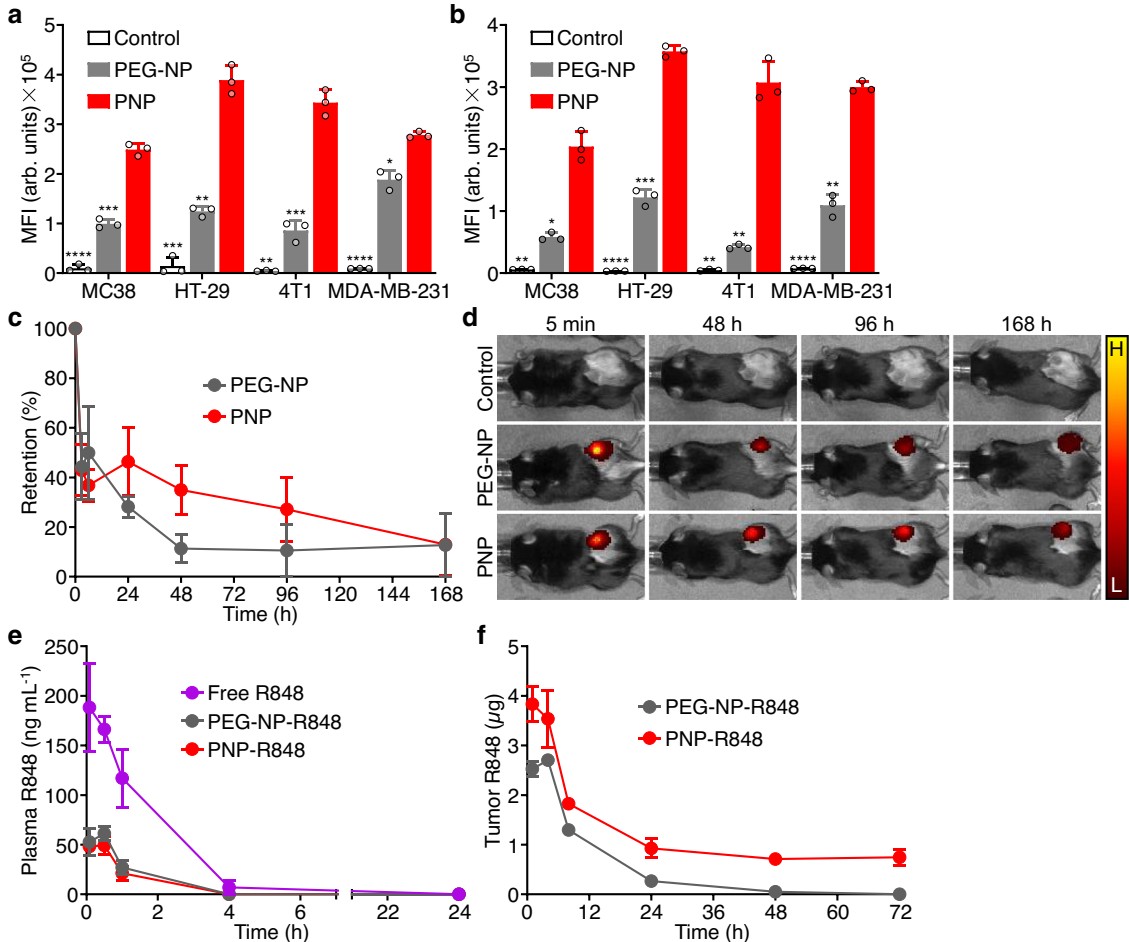

**Fig. 2 Nanoparticle interaction with tumor cells. a, b** Quantification of binding (**a**) and uptake (**b**) of PEG-NP and PNP by various cancer cells (MC38, HT-29, 4T1, and MDA-MB-231) after incubation in vitro ($n = 3$, mean + SD; MFI = mean fluorescence intensity). *$p < 0.05$, **$p < 0.01$, ***$p < 0.001$, ****$p < 0.0001$ (compared with PNP); two-way ANOVA. **c** Retention of PEG-NP or PNP over time after intratumoral administration into mice bearing MC38 tumors ($n = 3$, mean ± SEM). **d** Representative images from the study in (**c**) at 5 min, 48 h, 96 h, and 168 h (H = high fluorescent signal, L = low fluorescent signal). **e** Plasma levels of R848 after intratumoral administration of free R848, PEG-NP-R848, and PNP-R848 into mice bearing MC38 tumors ($n = 3$, mean ± SEM). **f** Retention of R848 after intratumoral administration of PEG-NP-R848 and PNP-R848 into mice bearing MC38 tumors ($n = 3$, mean ± SEM). Source data are provided as a Source Data file.

assays (ELISAs). For each cytokine that was studied, our results showed a dose-dependent release pattern that was similar for both samples. It was also confirmed that empty PNP, regardless of whether the platelet membrane was sourced from humans or mice, did not induce appreciable APC maturation or cytokine secretion (Supplementary Fig. 2); this supports the notion that the immune response elicited by PNP-R848 was driven largely by inclusion of the R848 payload.

**Nanoparticle interaction with immune cells**. We next sought to evaluate the interactions of the PNP formulation with various BMDC subpopulations (Fig. 3h, i). PNP showed a significant increase in both cell binding and uptake as compared to PEG-NP for all cell subtypes examined, including CD45+ leukocytes, CD11b+ macrophages, and CD11c+ dendritic cells. It is believed that the enhanced uptake of PNP by BMDCs may have contributed to the increased cytokine release observed in the preceding study. We next studied the in vivo interaction of the nano-formulation with tumor cell populations at various timepoints after intratumoral administration of dye-labeled PEG-NP and PNP (Fig. 3j–l). Overall, the uptake of PNP by the total population of cells in the tumor was significantly higher, evidenced by a

significant increase in fluorescence intensity compared to PEG-NP. When immune cell subsets in the tumor were evaluated, higher uptake for PNP was also observed among CD45+ leukocytes and CD11c+ dendritic cells across all timepoints.

**Antitumor efficacy in a mouse colorectal cancer model**. We evaluated the antitumor efficacy of PNP-R848 using an MC38 murine colon adenocarcinoma model in immunocompetent C57BL/6 mice (Fig. 4a). Each animal received a subcutaneous injection of $1 \times 10^6$ MC38 cells in the right flank, and the average tumor size was allowed to reach ~30–40 mm³. At this point, the mice started receiving one of the following treatments: 8% sucrose as a negative control, free R848, PEG-NP loaded with R848 (PEG-NP-R848), or PNP-R848, each at a drug dosage of 15 μg per injection. Treatments were administered intratumorally every other day for a total of 3 times, after which the mice were monitored regularly to assess therapeutic efficacy (Fig. 4b–e). Rapid regression followed PNP-R848 treatment, and complete tumor eradication was observed for 100% of the mice. Tumor growth was considerably delayed when treating with either free R848 or PEG-NP-R848, but there was significant disease progression in a majority of the mice by

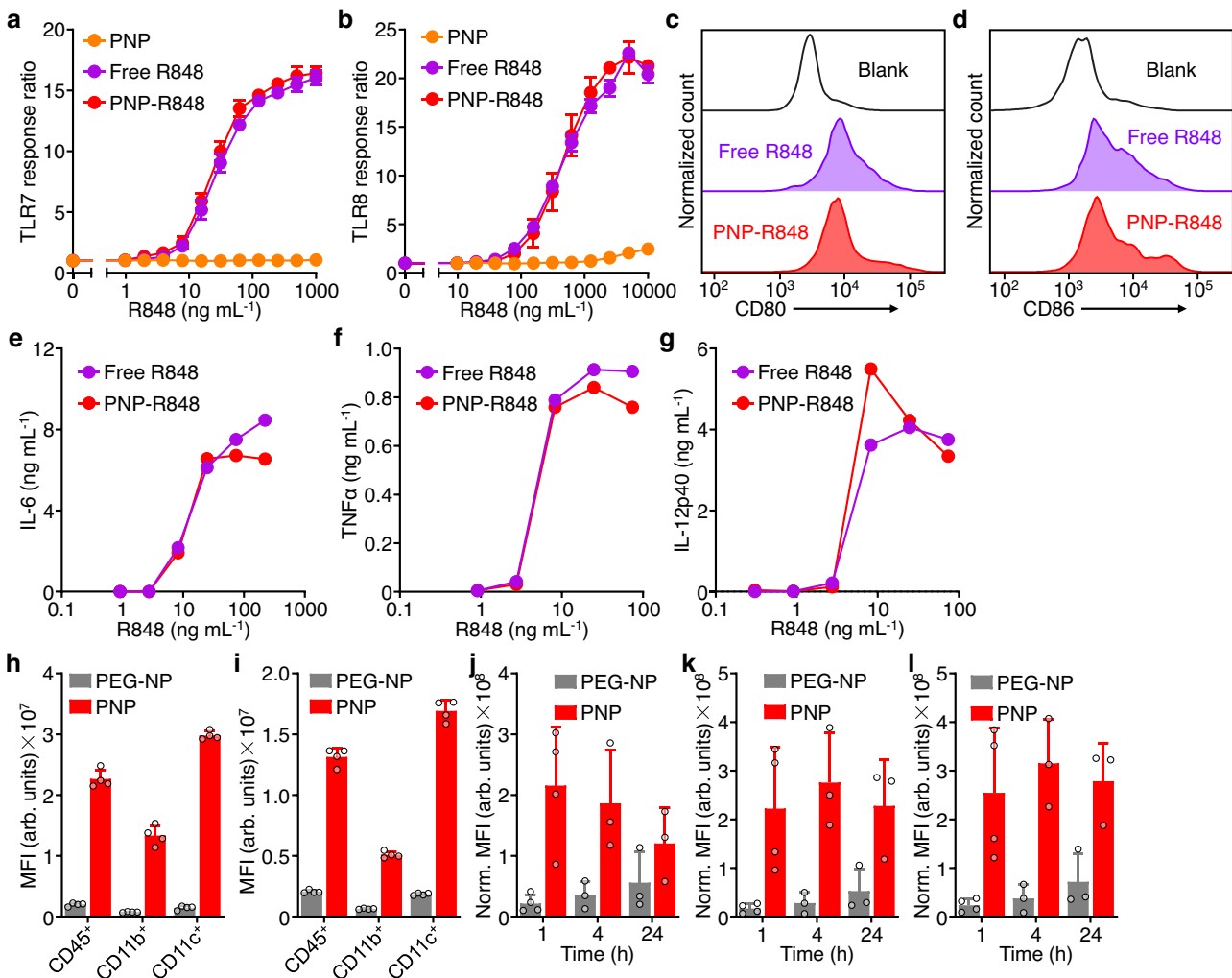

**Fig. 3 Nanoparticle in vitro activity and interaction with immune cells. a, b** Dose-dependent response of TLR7 (**a**) and TLR8 (**b**) reporter cell lines after incubation with PNP, free R848, and PNP-R848 ($n = 6$, mean + SD). **c, d** Expression of CD80 (**c**) and CD86 (**d**) by bone marrow-derived cells (BMDCs) after incubation with free R848 or PNP-R848. **e**–**g** Dose-dependent secretion of IL-6 (**e**), TNFα (**f**), and IL-12p40 (**g**) by BMDCs after incubation with free R848 and PNP-R848 ($n = 1$ for (**e**) and $n = 2$ for (**f** and **g**)). **h, i** Quantification of binding (**h**) and uptake (**i**) of PEG-NP and PNP by immune cell subsets (CD45$^+$, CD11b$^+$, and CD11c$^+$) after incubation with BMDCs in vitro ($n = 4$, mean + SD; MFI = mean fluorescence intensity). **j**–**l** In vivo uptake of PEG-NP and PNP by the total tumor cell population (**j**), CD45$^+$ cells (**k**), and CD11c$^+$ cells (**l**) at various timepoints after intratumoral administration ($n = 4$ for 1 h timepoint and $n = 3$ for rest of the timepoints, mean + SD). MFI was normalized based on the total cell number. Source data are provided as a Source Data file.

approximately 30 days after the start of treatment. In the end, both free R848 and PEG-NP-R848 treatments yielded a 28.6% long-term survival rate. We also evaluated treatment efficacy when reducing the drug dosage by 2.5-fold to 6 μg of R848 per injection (Supplementary Fig. 3). In this case 87.5% of mice treated with PNP-R848 completely rejected the tumor challenge, and 28.6% of mice survived after treatment with PEG-NP-R848. Interestingly, free R848 at the lower dosage outperformed the corresponding higher dosage treatment, with a 62.5% long-term survival rate. None of the treatments had a significant impact on the weight of the mice, suggesting that there was no acute toxicity. While there was a transient elevation of serum cytokine levels after treatment with PNP-R848, the values all returned to baseline within 48 h (Supplementary Fig. 4). It should also be noted that both PEG-NP and PNP without any R848 loading had minimal effect on the progression-free survival of tumor-bearing mice, and any differences that were observed compared with the control group were statistically insignificant (Supplementary Fig. 5).

In order to determine if the surviving animals had developed long-term immunity against MC38 cancer cells, the mice were re-challenged with a 3-fold higher inoculum implanted subcutaneously into the right flank 56 days after initiation of the first treatment (Fig. 4c, d). For the survivors that had been treated with either dose of PNP-R848, the second tumor challenge was rejected at a 100% rate. Though animals treated with the 6-μg dosage of free R848 initially demonstrated a 62.5% survival rate, the overall survival dropped to 37.5% after the tumor re-challenge, indicating inefficient development of an adaptive immune response against MC38 cells. The remainder of the surviving animals in the other groups all rejected the re-challenge, with no tumor progression observed at least 100 days after the start of the initial treatment. These results show that, while free R848 exhibits antitumor activity, it is not as efficient as the PNP-R848 formulation at eliciting long-lasting immunity. Notably, mice that were treated with PNP-R848 all rejected a second re-challenge with a 5-fold higher dose of cancer cells performed 140 days after initial treatment (Fig. 4c, d).

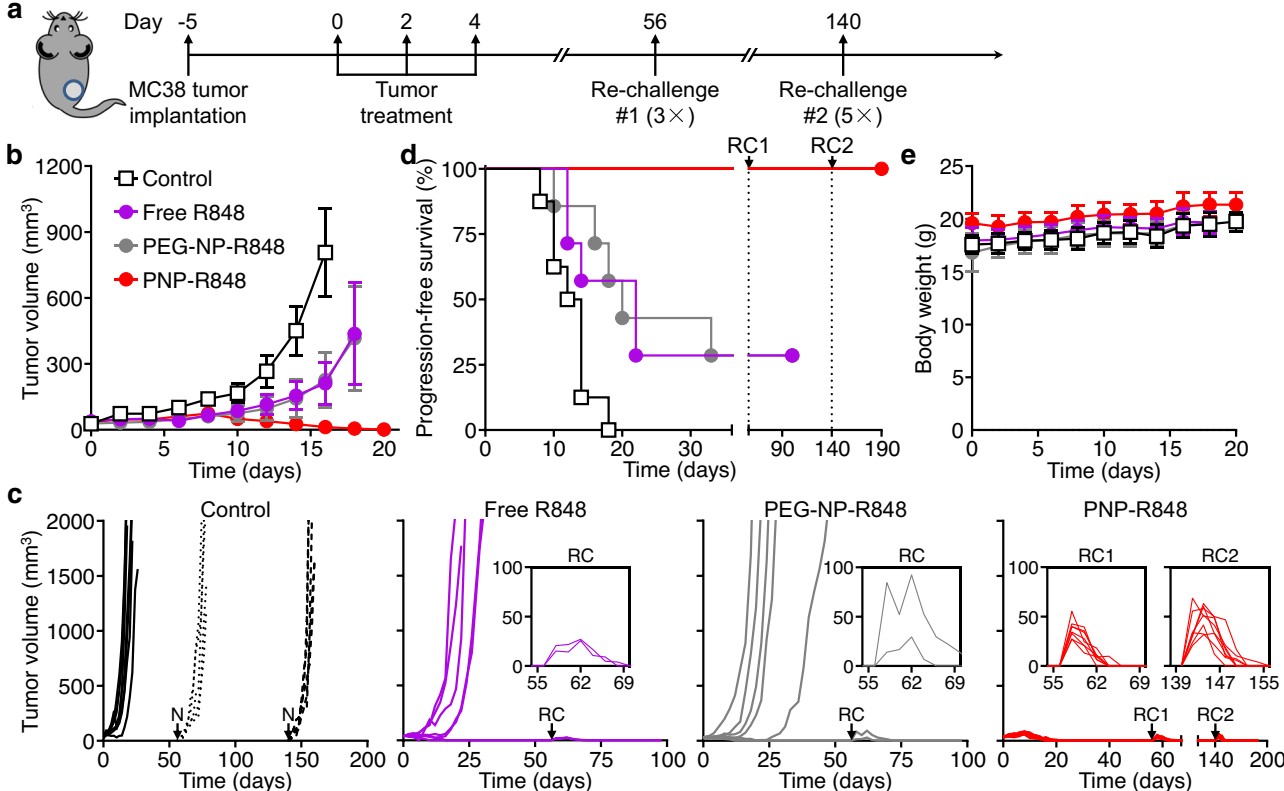

**Fig. 4 Therapeutic antitumor efficacy in an MC38 tumor model. a** Schematic timeline of the efficacy study. Tumors were treated with a sucrose control ($n = 7$), free R848 ($n = 7$), PEG-NP-R848 ($n = 7$), and PNP-R848 ($n = 8$). **b** Average tumor growth kinetics after treatment (mean ± SEM). **c** Individual tumor growth kinetics after treatment (N = naive challenge, RC = re-challenge). The insets depict the growth kinetics after each re-challenge. **d** Progression-free survival (tumor size < 200 mm$^3$) of mice after treatment. **e** Body weight of mice after treatment (mean + SD). Source data are provided as a Source Data file.

We also assessed the therapeutic efficacy of PNP-R848 in combination with chemotherapy (Supplementary Fig. 6). While free doxorubicin administered intratumorally at a high dosage of 63 μg prolonged the survival, the improvement was modest compared with PNP-R848 treatment. When combining the two treatment modalities, 100% of the mice survived the initial tumor challenge, although a greater than 10% loss in body weight 6 days after the start of treatment suggested the presence of toxicity. Despite some promising initial results, all mice treated with doxorubicin, both with or without PNP-R848, succumbed after re-challenge with a 3-fold higher dose of cancer cells. Given that leukodepletion is often a side effect of chemotherapy[53], these results highlight the need for an intact immune response to achieve long-lasting antitumor protection.

**Effect of treatment on immune cell populations in vivo.** To elucidate the immune responses associated with treatment efficacy, the DLN from tumor-bearing mice were collected 7 days after administration of low-dose free R848 or PNP-R848 on the same schedule as above. We found that PNP-R848 was able to significantly elevate the expression of major histocompatibility complex II (MHC-II), a maturation marker, on CD11b$^+$ and CD11c$^+$ APC subsets (Fig. 5a). No significant differences in MHC-II expression were observed in the same cell populations after free R848 treatment. Interestingly, the overall percentage of CD3$^+$ T cells in the DLN on day 7 dropped in response to PNP-R848 treatment (Fig. 5b), and this also held true for the proportion of CD8$^+$ T cells (Fig. 5c). Among the T cells that were present, the CD4$^+$ population had a significantly elevated proportion with the effector memory (CD44$^{hi}$CD62L$^{low}$) and central

memory (CD44$^{hi}$CD62L$^{hi}$) phenotypes (Fig. 5d). Since we observed a drop in the percentage of T cells in the DLN, we next assessed whether this was due to their migration into the tumor. The tumor tissue was histologically sectioned and stained for various immune cell subsets (Fig. 5e, f). Indeed, increased densities of both CD4$^+$ and CD8$^+$ T cells were found in the tumors of mice treated with PNP-R848 compared with free R848. Overall, our data indicate that PNP-R848, by improving tissue retention, was able to enhance the stimulation of APCs in the DLN, resulting in better priming of T cells and their subsequent recruitment into the tumor. This ultimately led to tumor eradication and the generation of memory T cells to fight subsequent tumor re-challenge.

**Antitumor efficacy in a mouse breast cancer model.** To further evaluate the applicability of PNP-R848 as a generalizable treatment against solid tumors, anticancer efficacy was tested in a syngeneic murine 4T1 triple-negative breast cancer model established using BALB/c mice (Fig. 6a). Each animal was subcutaneously implanted with $5 \times 10^5$ tumor cells in the right flank, and the average tumor size was allowed to reach ~30–40 mm$^3$ before treatment with either 8% sucrose, free R848, PEG-NP-R848, or PNP-R848 at a drug dosage of 15 μg per injection. The mice were treated every other day for a total of 5 times, and the tumor sizes and progression-free survival were monitored (Fig. 6b–d). Similar to the MC38 model, administration of PNP-R848 resulted in significant inhibition of 4T1 tumor growth. With PNP-R848 treatment, progression-free survival was prolonged to 23 days, compared to 9 days for the control group. Both free R848 and PEG-NP-R848 exhibited an intermediate level of antitumor

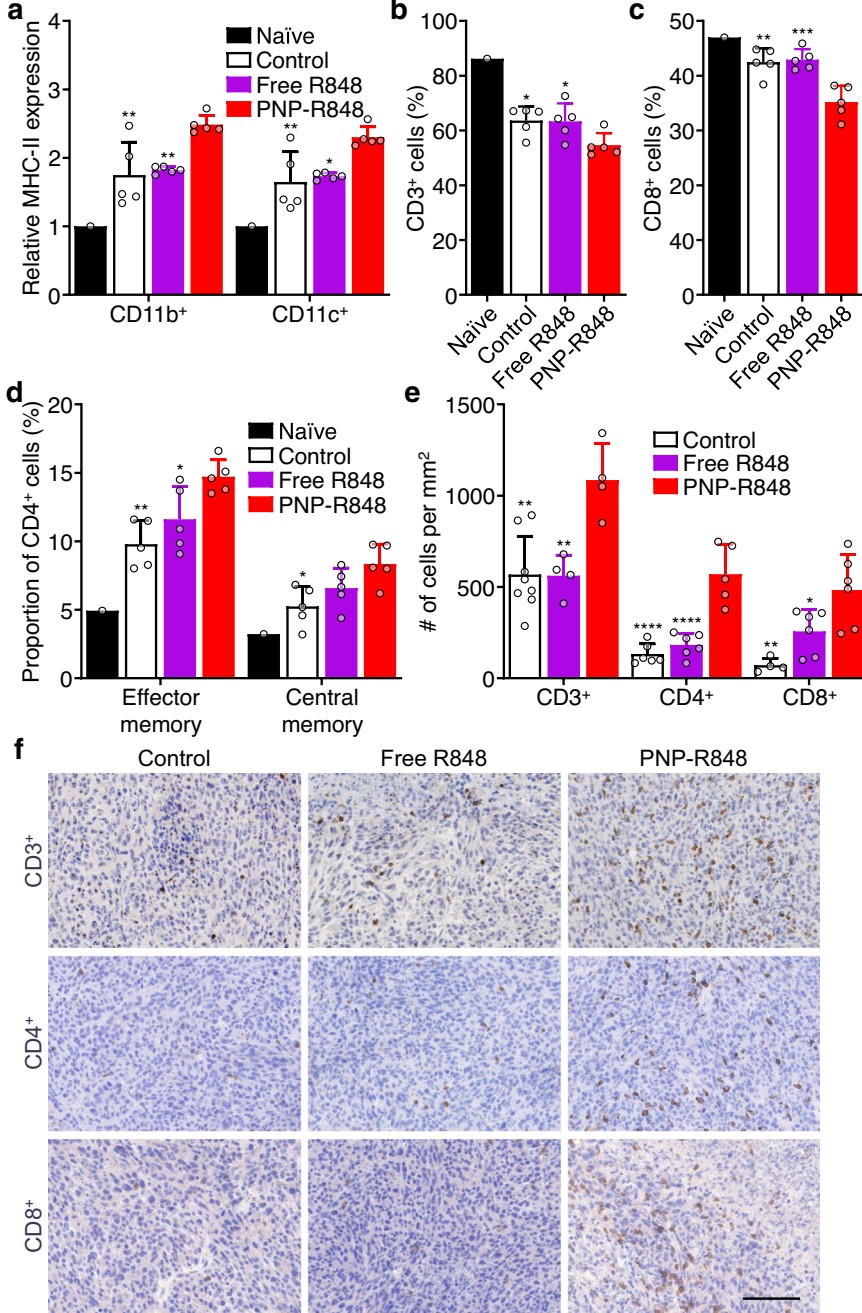

**Fig. 5 Immune response to treatment in MC38 tumor-bearing mice. a** Relative expression of MHC-II by CD11b+ or CD11c+ cells in the DLN from mice treated with free R848 and PNP-R848 ($n = 5$, mean + SD). **b** Percentage of CD3+ cells within the CD45+ cell population of the DLN from mice treated with free R848 and PNP-R848 ($n = 5$, mean + SD). **c** Percentage of CD8+ cells within the CD3+ cell population of the DLN from mice treated with free R848 and PNP-R848 ($n = 5$, mean + SD). **d** Proportion of CD4+ T cells with the effector memory or central memory phenotypes in the DLN from mice treated with free R848 and PNP-R848 ($n = 5$, mean + SD). **e** Quantification of CD3+, CD4+, or CD8+ cell density in tumor sections from mice treated with free R848 and PNP-R848 ($n = 4$ for CD3+ Free R848, CD3+ PNP-R848, and CD8+ Control, $n = 5$ for CD4+ PNP-R848, $n = 6$ for CD4+ Control, CD4+ Free R848, CD8+ Free R848, and CD8+ PNP-R848, and $n = 8$ for CD3+ Control, mean + SD). **f** Representative histological sections from the experiment in (**e**) (scale bar = 100 μm; brown = positive staining). All mice were treated on the same schedule as in Fig. 4, and samples were collected on day 7 after the first treatment. $*p < 0.05$, $**p < 0.01$, $***p < 0.001$, $****p < 0.0001$ (compared with PNP-R848); one-way ANOVA. Source data are provided as a Source Data file.

efficacy. This trend was also reflected on day 30 after the first treatment, when the tumors were excised and weighed (Fig. 6e, f). Notably, PNP-R848 had a marked effect on the number of metastatic nodules in the lungs, reducing the average number per lung to 3 nodules from more than 50 for the control group (Fig. 6g).

## Discussion

Here, we report on a biomimetic delivery vehicle to locally retain a potent immunomodulator at tumor sites. The TLR7 family of agonists stimulates dendritic cell activation and subsequent T cell priming, which leads to tumor-specific T cell immune responses and immunity[33,54,55]. There are some reports demonstrating that

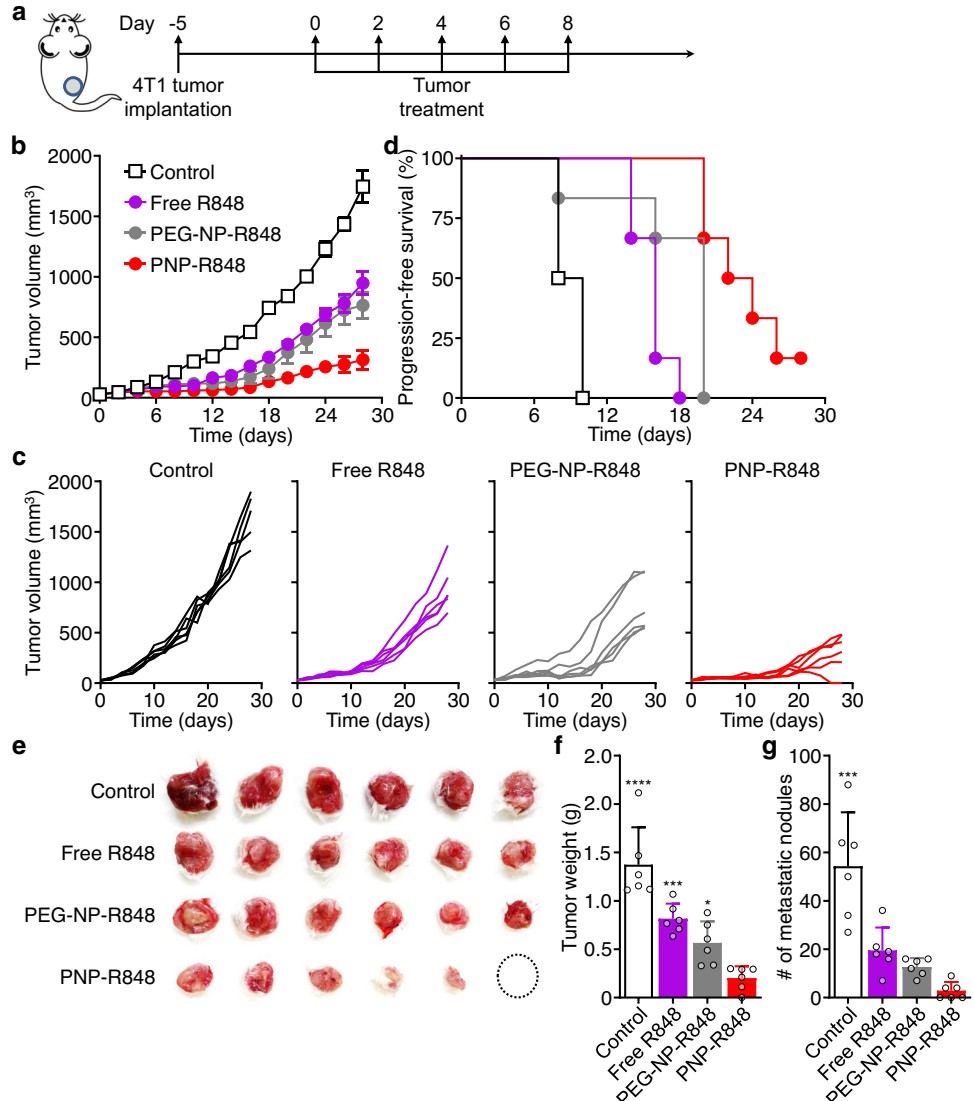

**Fig. 6 Therapeutic antitumor efficacy in a 4T1 tumor model. a** Schematic timeline of the efficacy study. Tumors were treated with free R848, PEG-NP-R848, or PNP-R848 ($n = 6$). **b** Average tumor growth kinetics after treatment (mean ± SEM). **c** Individual tumor growth kinetics after treatment. **d** Progression-free survival (tumor size < 200 mm³) of mice after treatment. **e** Images of tumors at day 30 post-treatment. **f** Average tumor weights at day 30 post-treatment (mean + SD). **g** Number of metastatic nodules in the lungs at day 30 post-treatment ($n = 6$, mean + SD). *$p < 0.05$, ***$p < 0.001$, ****$p < 0.0001$ (compared with PNP-R848); one-way ANOVA. Source data are provided as a Source Data file.

systemic administration of R848 via the intravenous or intraperitoneal route promotes antitumor immune responses, but high drug dosages are generally required to achieve therapeutic efficacy[18,56]. In one case, mice bearing MC38 tumors were administered with a total of 600 μg of R848, yet complete tumor regression was not observed[56]. In contrast, it has been shown that local delivery of R848 at more modest dosages, sometimes in combination with chemotherapy, can lead to complete tumor regression and long-term protective immunity[57,58]. It should be noted that considerable efficacy has generally been observed for intratumoral R848 only when combined with other immunostimulatory agents[58,59]. Our data suggest that the biomimetic platelet-derived membrane increased the interaction of PNP-R848 with various cells in the tumor microenvironment, thereby enhancing the bioavailability of R848 at the tumor site and surrounding lymphoid tissue after local delivery. This enabled us to significantly reduce the required dose of R848 compared to previous studies while maintaining its therapeutic potential. Even at a low total dosage of 18 μg per mouse, complete remission was

observed in almost all mice receiving PNP-R848 alone in an MC38 colorectal tumor model.

We hypothesize that PNP-R848 administered intratumorally is able to affect tumor regression by triggering a local inflammatory response and activating the resident APCs, some of which can migrate to the DLN and promote a subsequent influx of primed T cells into the tumor tissue. We observed the infiltration of cytotoxic CD8+ cells into the tumor as early as 7 days post-treatment, which corresponded to the beginning of tumor size reduction in the efficacy study. Our data corroborate recent findings that intratumoral activation of TLR7/8 transforms the tumor microenvironment and induces immune cell infiltration into the tumor[32]. Additionally, the observed increase in effector and central memory T cells in the DLN supports the development of systemic adaptive antitumor immunity. When re-challenged with more aggressive tumor implantation protocols, the animals that had eradicated the initial MC38 tumors following intratumoral treatments with PNP-R848 exhibited strong immunity and completely rejected the new implants within 2 weeks. Notably, we

also observed significant reduction in lung metastasis in a 4T1 breast cancer model after PNP-R848 treatment.

In conclusion, we have developed a biomimetic nanoformulation leveraging platelet membrane coating to enhance the delivery and retention of an immunostimulatory payload for intratumoral cancer immunotherapy. The membrane-coated nanoparticles efficiently interacted with cancer cells, leading to enhanced tumor retention in vivo and maximizing the activity of the encapsulated R848 payload. In an immunocompetent murine model of colorectal cancer, treatment with PNP-R848 was able to completely eradicate tumor growth, leading to long-term anti-tumor immunity that allowed all of the surviving mice to reject a subsequent re-challenge. The potent activity of the formulation was further corroborated in a murine model of triple-negative breast cancer, where significant reduction in metastasis was achieved. We believe that this approach for the localized delivery of small-molecule immunomodulators could be easily applied across a wide range of solid tumor types, providing a meaningful strategy for eliciting potent immune responses that could greatly enhance patient outcomes in the clinic.

## Methods

**Platelet membrane preparation and characterization.** Human platelet-rich plasma (PRP) was purchased from the San Diego Blood Bank. The PRP was collected from voluntary donors by the Blood Bank following all relevant ethical regulations. No identifying information was associated with purchased units of PRP upon our receipt. To collect the platelet membrane, the PRP was first diluted 2× with a buffer consisting of 140 mM NaCl (Fisher Chemical), 2.7 mM KCl (Fisher Chemical), 3.8 mM 4-(2-hydroxyethyl)-1-piperazineethanesulfonic acid (HEPES; Acros), 5 mM ethylene glycol-bis(β-aminoethyl ether)-N,N,N′,N′-tetra-acetic acid (Bioworld), and 2 μM prostaglandin E1 (PGE1; AdooQ BioScience), followed by centrifugation at 2000g for 15 min with no brake. The supernatant was removed and the platelets were resuspended with a lysis buffer containing a mixture of 75 mM NaCl, 6 mM NaHCO₃ (Fisher Chemical), 1.5 mM KCl, 0.17 mM Na₂HPO₄ (Fisher Chemical), 0.5 mM MgCl₂ (Alfa Aesar), 20 mM HEPES, 1 mM ethylenediaminetetraacetic acid (Fisher Chemical), 1 μM PGE1, 0.01% NP40 surfactant (Boston Bioproducts), and protease inhibitors (Thermo Scientific). The platelet membrane was derived by a repeated freeze-thaw process. The platelet mixture was frozen at −80 °C, thawed at room temperature, and pelleted by centrifugation at 21,100g for 10 min. The pellet was then resuspended in the lysis buffer, and the freeze-thaw was repeated two more times. After the repeated washes, the membrane was suspended in water for coating onto the nanoparticle cores.

Quantification of total membrane protein concentration was performed using a Pierce BCA protein assay kit (Life Technologies). Flow cytometry was used to probe for the expression of specific surface markers on the platelet membrane using FITC-conjugated annexin V (Biolegend), Alexa488-conjugated anti-human P-selectin (AK4; Biolegend), Alexa647-conjugated anti-human GPIbα (HIP1; Biolegend), and Alexa647-conjugated anti-human αIIbβ3 (PAC-1; Biolegend). The probes (1:200 dilution) were incubated with purified platelet membrane in PBS (Gibco) for 30 min in the dark at room temperature. After incubation, the membrane was washed by centrifugation at 21,100g. Data were collected using a Becton Dickinson Accuri C6 flow cytometer equipped with BD Accuri C6 Plus software and analyzed with FlowJo V10 software.

**Nanoparticle synthesis and physicochemical characterization.** R848-loaded nanoparticles were synthesized using a single emulsion process. First, PLA (R202H; Evonik) and R848 (BOC Sciences) were dissolved in an organic phase consisting of benzyl alcohol (Acros) and ethyl acetate (Fisher Chemical) at concentrations of 60 and 10 mg mL⁻¹, respectively. The mixture was then added to 5× volume of ice-cold outer phase media consisting of 10 mM Tris pH 7.5 (Invitrogen) with 0.2 wt% sodium cholate (Alfa Aesar) and 7 vol% ethyl acetate. This solution was homogenized at 12,000 rpm for 90 s using a Kinematica Polytron PT 3100 homogenizer before being passed through a Microfluidics LM20 Microfluidizer (outfitted with a Y chamber) 3 times. This mixture was then added to an equal volume of outer phase media, and the solvent was evaporated overnight in a fume hood while stirring at 200 rpm. Unloaded nanoparticle cores were fabricated using the same procedure without R848 in the organic phase. Platelet membrane coating was performed by sonication of the R848-loaded or unloaded nanoparticle cores with platelet membrane at a polymer-to-platelet membrane protein mass ratio of 1:0.7 for 2 min in a Branson CPX3800H ultrasonic bath at a frequency of 40 kHz. This process is expected to lyse any intact platelet vesicles present in the membrane preparation. PEG-coated nanoparticles were fabricated using the same procedure as for the nanoparticle cores, but using PEG-conjugated PLA (PolySciTech) to replace 10 wt% of the unconjugated PLA. To prepare nanoparticles loaded with

1,1′-dioctadecyl-3,3,3′,3′-tetramethylindodicarbocyanine (DiD; Biotium), the dye was added to a 10 mg mL⁻¹ PLA solution in acetone at 0.1 wt% of the polymer. Then, 2 mL of this solution was added dropwise to 4 mL of water to form the dye-loaded nanoparticle cores. After overnight solvent evaporation, the nanoparticles were coated with platelet membrane by sonication. Hydrodynamic nanoparticle size and surface zeta potential were measured by dynamic light scattering using a Malvern Zetasizer Nano ZS. For imaging, the nanoparticles were stained with 0.2 wt% uranyl acetate (Electron Microscopy Sciences) and visualized with an FEI Tecnai Spirit G2 BioTWIN transmission electron microscope. For stability characterization, uncoated NP-R848 or coated PNP-R848 were suspended at 1 mg mL⁻¹ in PBS and stored either at room temperature or at 4 °C. At weekly intervals, the size of the samples was measured by dynamic light scattering using a Malvern Zetasizer Nano ZS. All nanoformulations were passed through a 0.2-μm filter for sterilization prior to in vitro and in vivo experiments.

**Quantification of platelet activation factors.** PRP, platelet lysate, and purified platelet membrane were prepared and examined for the platelet-activating molecules thrombin and ADP using a SensoLyte 520 thrombin activity assay kit (Anaspec) and a PicoProbe ADP assay kit (BioVision), respectively, based on the manufacturers' instructions. To prepare the PRP, lysate, and platelet membrane samples, 1 mL of PRP was first diluted 3× and divided into 3 groups. The first group was used directly, the second group was washed and underwent one freeze-thaw cycle to produce the platelet lysate, and the third group was processed to completion into purified membrane. All samples were resuspended to the same volume with PBS prior to performing the assays.

**Drug loading and release.** R848 loading was analyzed using a reverse-phase ultra-high-performance liquid chromatography (UHPLC) method. The UHPLC system consisted of a binary gradient pump, in-line degasser, autosampler and Thermo Scientific Vanquish photodiode array detector. Separation and quantitative analysis of R848 were achieved on a 3.5 μm Waters XBridge™ C18 column (2.1 × 150 mm) with the mobile phase flowing at a rate of 1.0 mL min⁻¹ and a detection wavelength of 227 nm. Mobile phase A consisted of 10 mM sodium phosphate (Fisher Chemical) with 0.1% triethylamine (Acros) and pH adjusted to 2.45, while mobile phase B consisted of 100% acetonitrile (Fisher Chemical). The acquisition run time for each analysis was 6.5 min with a gradient consisting of 15% mobile phase B from 0 to 3 min, 45% mobile phase B from 3 to 5 min, and 15% mobile phase B from 5.1 to 6.5 min. The samples were first diluted in acetonitrile and then diluted in a combination of 30% acetonitrile and 70% 0.1N hydrochloric acid (Acros). They were then injected into the column after a series of six standard injections prepared by diluting R848 in 100% acetonitrile. Drug release kinetics from PNP-R848 were evaluated utilizing 20 kDa dialysis cassettes (Thermo Scientific). Reconstituted samples were transferred to the cassettes via a syringe with a 21-gauge needle and dialyzed against a large volume of PBS. The dissolution experiments were run at room temperature while stirring the PBS solution at 280 rpm for 6 days. Samples were collected at various timepoints (30 min, 1 h, 2 h, 4 h, 6 h, 8 h, 1 day, 2 days, 3 days, and 6 days) and analyzed by UHPLC.

**In vitro binding and uptake of PNP by murine and human cancer cells.** MC38 murine colon adenocarcinoma cells (Kerafast) and MDA-MB-231 human mammary gland adenocarcinoma cells (HTB-26; American Type Culture Collection) were cultured in Dulbecco's modified Eagle's medium (Gibco) supplemented with 10% fetal bovine serum (Corning). 4T1 murine mammary gland cancer cells (CRL-2539; American Type Culture Collection) were cultured in RPMI-1640 medium (Gibco) supplemented with 10% fetal bovine serum. HT-29 human colorectal adenocarcinoma cells (HTB-38; American Type Culture Collection) were cultured in McCoy's 5a medium (Gibco) supplemented with 10% fetal bovine serum. For the binding study, DiD-loaded PNP or PEG-NP were incubated with 5 × 10⁵ MC38, HT-29, 4T1, or MDA-MB-231 cells in 100 μL of media. The final nanoparticle concentration for this incubation was 0.2 mg mL⁻¹. The incubation was performed for 30 min at 4 °C in order to minimize endocytic uptake, after which the cells were washed 3 times with PBS and examined using flow cytometry. For the uptake study, the incubation was instead performed for 10 min at 37 °C. Data were collected using a Becton Dickinson Accuri C6 flow cytometer equipped with BD Accuri C6 Plus software and analyzed with FlowJo V10 software.

**TLR activation assays.** HEK-Blue hTLR7 and HEK-Blue hTLR8 reporter cells (Invivogen) were cultured as directed by the manufacturer. For the dose–response experiments, 20 μL of PNP, free R848, or PNP-R848 were loaded into 96-well cell culture plates at 10× the desired final concentration (0.977 to 1000 ng mL⁻¹). The cultured reporter cells were rinsed with warm PBS, resuspended in 1 mL warm PBS, and then detached from culture flasks by gentle scraping. The cells were diluted to a concentration of 2.2 × 10⁵ cells mL⁻¹ in HEK-Blue detection medium (Invivogen), whereupon 180 μL of the cell suspension was immediately added to the sample dilutions. Absorbance at 655 nm was measured after 21 h of incubation at 37 °C in 5% CO₂.

**Nanoparticle activity on BMDCs**. All animal experiments were performed in accordance with guidelines approved by the Institutional Animal Care and Use Committee of the University of California San Diego. Animals were maintained in standard housing at 68–75 °F, 40–60% relative humidity, and 12 h light/dark cycles. Female C57BL/6 mice (Charles River Laboratories) were euthanized via $CO_2$ asphyxiation. Intact tibias were isolated from each mouse, dipped briefly into 70% ethanol, and stored in RPMI cell culture media (Gibco) over ice. Both ends of each tibia were cut and each bone was flushed with 10 mL of RPMI using a syringe attached with a 23-gauge needle. Bone marrow cells were collected and washed by centrifugation at 320$g$ for 9 min. Finally, cells were passed through a 50-μm cell strainer (Corning). For cytokine release and co-stimulatory marker characterization, BMDC cells were counted, and $5 \times 10^5$ cells were plated per well in 6-well plates. Various concentrations of free R848 and PNP-R848 were added to the cells and incubated at 37 °C for 24 h. Afterwards, the supernatant was assayed using ELISA kits for IL-6 (BD Biosciences), TNFα (R&D Systems), and IL-12p40 (R&D Systems). The cells were washed and scraped from the plates, followed by staining with FITC-conjugated anti-mouse CD45 (30-F11; BD Biosciences; 1:400 dilution), PE-conjugated anti-mouse CD80 (16-10A1; BD Biosciences; 1:400 dilution), and APC-conjugated anti-mouse CD86 (GL-1; Biolegend; 1:400 dilution). Data were collected using a Becton Dickinson Accuri C6 flow cytometer equipped with BD Accuri C6 Plus software and analyzed with FlowJo V10 software (Supplementary Fig. 7). In a separate experiment, empty PNP fabricated from mouse platelets (mPNP) or human platelets (hPNP), free R848, or PNP-R848 were added to the cells at an R848 concentration of 24.6 ng mL$^{-1}$ (or equivalent nanoparticle amount) and incubated at 37 °C for 24 h. Lipopolysaccharide (LPS; Invivogen) at 20 ng mL$^{-1}$ was used as a positive control. Afterwards, the supernatant and cells were analyzed as described above.

**BMDC binding and uptake**. For cell binding studies, DiD-loaded PNP or PEG-NP were incubated with $1 \times 10^6$ BMDC cells in 100 μL media at a final nanoparticle concentration of 0.2 mg mL$^{-1}$. Incubation was performed for 30 min at 4 °C, after which the cells were washed 3 times with PBS and subsequently stained using FITC-conjugated anti-mouse CD45 (30-F11; BD Biosciences; 1:400 dilution), PE-conjugated anti-mouse CD11b (M1/70; Biolegend; 1:400 dilution), and PE/Cy7-conjugated anti-mouse CD11c (N418; Biolegend; 1:400 dilution). For uptake studies, incubation was performed for 10 min at 37 °C. Data were collected using a Becton Dickinson Accuri C6 flow cytometer equipped with BD Accuri C6 Plus software and analyzed with FlowJo V10 software (Supplementary Fig. 7).

**In vivo interaction with tumors**. All animal experiments were performed in accordance with ethical regulations for animal testing and research and were approved by the Institutional Animal Care and Use Committee of the University of California San Diego. To develop tumors, $1 \times 10^6$ MC38 cells were implanted subcutaneously into the right flank of 6-week old female C57BL/6 mice. Tumor volumes were calculated using the equation: volume = (length × width$^2$)/2. For the tumor retention study, the average tumor size was allowed to reach 100 mm$^3$, after which the mice were administered with 8% sucrose as the negative control ($n = 3$), DiD-labeled PNP ($n = 3$), and DiD-labeled PEG-NP ($n = 3$). Each animal received one intratumoral injection and was imaged using a Xenogen IVIS 200 system at various timepoints, including 5 min, 3 h, 6 h, 24 h, 48 h, 96 h, and 168 h with the same acquisition time and filter settings. Acquired images were analyzed by the Xenogen Living Image software 3.0 to quantify the fluorescence intensity of the tumors and to determine tumor retention percentage.

To assess interaction with immune cells, mice with tumors with an average volume of 100 mm$^3$ were intratumorally administered with DiD-labeled PNP or PEG-NP. At 1 h, 4 h, and 24 h, groups of mice were euthanized, and the tumor tissue was processed into single-cell suspensions by digesting in a solution containing collagenase IV (Sigma-Aldrich) and DNase type IV (Sigma-Aldrich) at final concentrations of 1 mg mL$^{-1}$ and 10 μg mL$^{-1}$, respectively. The cells were stained using FITC-conjugated anti-mouse CD45 (1:400 dilution), PE/Cy7-conjugated anti-mouse CD11c (1:400 dilution), and LIVE/DEAD Fixable Aqua Stain (Invitrogen). Data were collected using a Becton Dickinson Accuri C6 flow cytometer equipped with BD Accuri C6 Plus software and analyzed with FlowJo V10 software (Supplementary Fig. 7).

**In vivo quantification of R848**. For serum analysis, $1 \times 10^6$ MC38 cells were implanted subcutaneously into the right flank of 6-week old female C57BL/6 mice. When the average tumor size reached 100 mm$^3$, mice were randomized into three groups and received a single intratumoral treatment of 8% sucrose ($n = 3$), PEG-NP-R848 ($n = 3$), and PNP-R848 ($n = 3$). Each group received 15 μg of R848. Blood samples were collected in BD Microtainer collection tubes at 5 min, 30 min, 1 h, 4 h, 24 h, 48 h, and 72 h post-treatment. The blood samples were centrifuged at 15,000$g$ for 2 min to separate the plasma, which was then analyzed using UHPLC for levels of R848. To assess R848 tumor retention, $1 \times 10^6$ MC38 cells were implanted by subcutaneous injection into the right and left flank of each C57BL/6 mouse. When the average tumor size reached 100 mm$^3$, mice were randomized into two groups and received a single intratumoral treatment of PEG-NP-R848 ($n = 12$) and PNP-R848 ($n = 12$) into each tumor. At 1 h, 4 h, 8 h, 24 h, 48 h, and 72 h

post-treatment, two mice from each group were euthanized, and their tumors were collected and processed for analysis of R848 levels by UHPLC.

**Therapeutic efficacy in a murine MC38 tumor model**. Mice were implanted with $1 \times 10^6$ MC38 cells subcutaneously into the right flank, which were allowed to grow to an average size of ~30–40 mm$^3$. The mice were then intratumorally treated every other day for a total of 3 times. The treatment groups included: 8% sucrose, free R848, PEG-NP-R848, and PNP-R848. Each group received 6 or 15 μg of R848 per treatment. The injection volume was 30 μL for all the treatments, and delivery was done via syringe with a 31-gauge needle. Care was taken to administer the solution slowly to prevent leakage. Tumor growth and mouse weight were monitored every other day. Progression-free survival was defined as tumor volume <200 mm$^3$, a threshold above which, based on our experience with the model, complete tumor regression is unlikely. All the mice that rejected the initial MC38 inoculum were re-challenged subcutaneously using $3 \times 10^6$ MC38 cells on day 56 after the start of the first treatment. Mice in the PNP-R848 treatment group that were tumor-free at the end of the initial re-challenge study were re-challenged a second time with $5 \times 10^6$ MC38 cells on day 140. For each re-challenge, 5 naive C57BL/6 mice that received the same MC38 tumor cell challenge were used as controls to verify tumorigenicity.

**Systemic cytokine release**. Mice were implanted with $1 \times 10^6$ MC38 cells subcutaneously into the right flank, which were allowed to grow to an average size of 150–200 mm$^3$. The mice received intratumoral injections of 8% sucrose and PNP-R848. Blood samples were collected in BD Microtainer collection tubes at 2 h, 8 h, 24 h, and 48 h post-treatment. The blood samples were centrifuged at 15,000$g$ for 2 min to separate the plasma, which was then analyzed using ELISA kits for IL-6, TNFα, and IL-12p40.

**Therapeutic efficacy with unloaded nanocarriers**. Mice were implanted with $1 \times 10^6$ MC38 cells subcutaneously into the right flank, which were allowed to grow to an average size of ~30–40 mm$^3$. The mice received intratumoral treatments every other day for a total of 4 times. The treatment groups included: 8% sucrose, PEG-NP, and PNP. The nanoparticles used in this study were empty and did not contain R848. Tumor growth and mouse weight were monitored every other day. Progression-free survival was defined as tumor volume <200 mm$^3$.

**Therapeutic efficacy in combination with doxorubicin**. Mice were implanted with $1 \times 10^6$ MC38 cells subcutaneously into the right flank, which were allowed to grow to an average size of ~30–40 mm$^3$. The mice received intratumoral treatments every other day for a total of 3 times. The treatment groups included: 8% sucrose, free doxorubicin, and doxorubicin + PNP-R848. Mice received 63 μg of doxorubicin and 15 μg of R848 per dose. For the combination treatment, doxorubicin and PNP-R848 were mixed pre-administration and the animal received one intratumoral injection containing both. Tumor growth and mouse weight were monitored every other day. Progression-free survival was defined as tumor volume <200 mm$^3$. All the mice that rejected the initial MC38 inoculum were re-challenged subcutaneously using $3 \times 10^6$ MC38 cells on day 56 after the start of the first treatment. Five naive C57BL/6 mice that received the same MC38 tumor cell challenge were used as controls to verify tumorigenicity.

**In vivo immune profiling**. Mice bearing MC38 tumors were treated on the same schedule as the antitumor efficacy study with 8% sucrose, free R848, and PNP-R848. Each group received 6 μg of R848. The mice were then euthanized 7 days after the first treatment, and the inguinal DLN (on the same side as the tumor) was processed into a single-cell suspension by shearing the tissue using a 50-μm cell strainer. Cells were stained with Fixable Viability Stain 450 (BD Biosciences) and different antibodies, including BV510-conjugated anti-mouse CD3 (17A2; Biolegend; 1:400 dilution), FITC-conjugated anti-mouse CD4 (RM4-5; eBiosciences; 1:400 dilution), APC/Cy7-conjugated anti-mouse CD8 (53-6.7; Invitrogen; 1:400 dilution), PerCP/Cy5.5-conjugated anti-mouse CD62L (MEL-14; eBioscience; 1:400 dilution), APC-conjugated anti-mouse CD44 (IM7; BD Biosciences; 1:400 dilution), V500-conjugated anti-mouse CD45 (30-F11; BD Biosciences; 1:400 dilution), APC-conjugated anti-mouse MHC-II (M5/114.15.2; Tonbo Bioscience; 1:400 dilution), APC/Cy7-conjugated anti-mouse CD11b (M1/70; BD Biosciences; 1:400 dilution), and PE/Cy7-conjugated anti-mouse CD11c (1:400 dilution). Data were collected using a Becton Dickinson FACSCanto II flow cytometer equipped with BD FACSDiva software and analyzed with FlowJo V10 software (Supplementary Fig. 7). Tumor tissues were fixed in formalin (Fisher Scientific) for 24 h and were then transferred into 70% ethanol prior to histological sectioning by the Moores Cancer Center Tissue Technology Shared Resource. Tumor sections were stained for mouse CD3, CD4, and CD8 using AEC substrate and counterstained with Mayer's Hematoxylin. Slides were imaged with a Hamamatsu Nanozoomer 2.0HT slide scanner and post-acquisition quantifications were made using QuPath v0.2.0 (open-source software from Github).

**Therapeutic efficacy in a murine 4T1 tumor model**. Female BALB/c mice (Charles River Laboratories) were implanted with $5 \times 10^5$ 4T1 cells subcutaneously into the right flank, which were allowed to grow to an average size of ~30–40 mm$^3$.

The mice were then treated every other day for a total of 5 treatments. The treatment groups included: 8% sucrose ($n = 6$), free R848 ($n = 6$), PEG-NP-R848 ($n = 6$), and PNP-R848 ($n = 6$). Each group received 15 μg of R848 per treatment. The injection volume was 30 μL for all the treatments, and delivery was done via syringe with a 31-gauge needle. Tumor growth and mouse weight were monitored every other day. Progression-free survival was defined as tumor volume <200 mm³. The study was terminated 30 days after the first treatment, and the tumors and lungs were harvested. For metastatic nodule counting, lung tissues were fixed using Bouin's solution (Electron Microscopy Sciences).

**Graph generation and statistical analysis**. GraphPad Prism 8 software was used for all graph generation and statistical analysis.

**Reporting summary**. Further information on research design is available in the Nature Research Reporting Summary linked to this article.

## Data availability

All data are available within the Article, Supplementary Information or available from the corresponding authors upon reasonable request. Source data are provided with this paper.

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

## Acknowledgements

We thank Alborz Amirsadeghi and Mao Yin for their help in the early development stage.

## Author contributions

B.B., H.G., R.H.F., and J. Zhang conceived and designed the experiments. B.B., H.G., B.T.L., K.J.H., E.D., M.P., J. Zhou, and W.G. performed the experiments. The manuscript was written by B.B., R.H.F., and J. Zhang. J.D.B. and L.Z. helped with study design and critically revised the manuscript. All authors discussed the results and reviewed the manuscript.

## Competing interests

These authors declare the following competing financial interests: B.B., B.T.L., K.J.H., E.D., M.P., and J. Zhang are employees of Cello Therapeutics. L.Z. has financial interest in Cello Therapeutics. The remaining authors declare no competing interest.
