## [Peer Review File. · Nature Communications]

REVIEWER COMMENTS

Reviewer #1 (Remarks to the Author): with expertise in platelet membranes and nanoparticles

In this manuscript, Bahmani et al. are attempting to exploit a new immunotherapeutic strategy against cancer by engaging TLRs in the tumor micro-environment and inhibit tumor-promoting immune signalling. They use resiquimod (R848) small molecule immunomodulator, as a model of TLR7/8 agonists suggested to induce an effective antitumor immune response. The systemic administration of R848 (or other TLR7 agonists) exerts toxic effect that limit clinical translation, leading to the exploration of intratumoral injection. As direct intratumoral injection of R848 may lead to drug leakage in the blood circulation, the authors evaluated the possibility of use platelet membrane-cloaked nanoparticle (PNP) for intratumoral delivery of R848, based on the scientific rationale that the numerous markers of the platelet membranes would be expected to lead to enhanced adherence and retention within the tumor site, thereby resulting in stronger anti-cancer efficacy upon intratumoral administration. To support their hypothesis the authors used a murine colorectal adenocarcinoma model and present data indicating that after intratumoral administration of the human platelet membrane coated PNP, there is an activation of APCs within the draining lymph node (DLN) and an increased immune infiltration. Further, they observed an efficient antitumoral response leading to short-term rejection of the induced tumors as well as long-term immunity tumor induction re-challenges.

Comments

The authors are asked to consider the following remarks:

Platelet membrane preparation and characterization: a more complete characterization of the platelet membrane for instance by DLS, TEM or SEM would be useful. The way used to prepare the membrane can very well lead to the formation of platelet extracellular vesicles which could "contaminate" the pellet during ultracentrifugation at 21'100 x g (even for 10 min only). Do the authors truly obtain isolated platelet membranes or was there "contamination" with extracellular vesicles and potential experimental interference for the preparation of the PNP? Please comment.

There is no data about drug loading efficacy of R848 in the NPs and little details about the loading methodology used which does not allow to reproduce the data. There are ample technical details about the (relatively standard) HPLC methodology to quantify R848...but by contrast essentially no details about what makes this engineering approach novel such as the coating methodology by sonication of the platelet membrane on the loaded or unloaded NP. Citing ref 37 (in the results section) may not be sufficient to be able to reproduce the technique. What are the conditions used for sonication? In addition, does the sonication affect the R-848 loading? What a polymer to membrane mass ratio of 1:0.7 means in non-equivocal terms? Does "the platelet membrane mass" means the protein content?

Why was the drug release kinetics of PNP-R848 was performed in the presence of 0.05% Triton X-100? I would strongly assume that this non-ionic detergent can artificially affect platelet membrane lipids, increase the permeability of the membrane and thus the release of R-848. This in itself could explain why the release "for both the bare NP-R848 and coated PNP-R848 formulations matched closely". Please clarify why Triton X-100 is used and demonstrate that it does not affect the release from PMP-R848.

Also it is much unclear how the 20 kDa dialysis cassettes was used. It seems to be an unusual manner to perform drug release experiments. Does it imply that the material was recirculated through the dialysis cassettes, as done for TFF? This may cause shear stress, speed-up the release, not mimicking well physiological situations. Please explain better how this is done and explain how this dialysis methodology was validated in your hands.

How were the various NPs sterilized for in vitro and in vivo experiments?

MC38 binding and uptake: the incubation was performed at 4 °C for determination of binding. Can the use of 4°C artificially affect the cells and impact the binding data?

A major issue that may invalidate the claimed data is the fact that the authors used human platelets to prepare the membrane coated NP and used C57BL/6 immunocompetent mice for the various explorations, including interaction with immune cells. To which extent are the noted impact on e.g. BMDC, activation of APCs within the DLN and, most critically, increased immune infiltration linked to the functional design of the NP, or simply to an inter-species human-mice immune reactivity (which would then make the system inefficient in humans using human platelets coated-NP-R848)? Also I did not see whether the BMDC are human or rodent cells. Was a validation showing similar effects done using NP coated with platelet membranes from mice performed by the author?

The statistical methods are not described. Please provide the missing information.

The presence of phosphatidylserine a potent pro-coagulant phospholipid on the platelet membranes seems to contradict the fact that assays for thrombin and adenosine diphosphate did not reveal a prothrombotic activity. How can this be explained? In addition, the precise experimental conditions used to carry these assays are not described in the method section. It should be, as well as the way how control materials (PRP, platelet lysate) were prepared. The evil is in the details. Please add experimental details e.g. as supplementary file.

The authors seem to administer a dose of 30 uL into 30 - 40 mm³ tumors. This is a very high dose for what are actually quite small tumors. Please comment in the discussion.

A progression-free survival defined as tumor volume < 200 mm³ seems to be a small value. Please clarify

Fig 1. G : as early kinetics of release is more important, a zoom on the 0-10 hours earlier period could be added (minor comment)

Fig 1.F Only 2 nanoparticles are shown. This does not give an idea of a whole population (e.g. 50-100) of NPs, in particular the risk of aggregation. In addition, comparative cryo-TEM pictures of the bared NP and PNP should be presented.

Fig 1.D The increase in size after coating is around 25 nm. Which brings the platelet membrane thickness to 12.5 nm. Does this corresponds to the actual platelet membrane thickness ?

Reviewer #2 (Remarks to the Author): with expertise in TLR agonists and immunotherapy

Background:

The authors of this manuscript designed an intratumoral (IT) platelet membrane-coated nanoparticle (PNP) to promote local localized delivery of the TLR agonist resiquimod (R848) with an aim to induce anti-tumour immunity effects whilst ameliorating systemic side-effects. They showed IT dosing of the PNP loaded with R848 (PNP-R848) induced tumor regression in a single syngeneic mouse cancer model, and animals with a complete response to PNP-R848 demonstrated immunity to tumor re-challenge. By flow cytometry and immunohistochemistry analysis of draining

lymph node (DLN) and tumor tissues(s) they describe enhanced immune activation in the PNP-R848 treatment arm versus a "more traditional" R848 formulation. Not much of the information provided in this manuscript is really novel or unexpected. In addition to the preclinical oncology studies referenced in this manuscript, describing encapsulation of adjuvants, such as TLR agonists, within nanoparticles (e.g. Schmid, D. et al Nat. Comms. (2018); Da Silva, CG. et al. Biomaterials (2019) there is a surfeit of other papers on this topic e.g. Qianqian, N. et al. "A bi-adjuvant nanovaccine that potentiates immunogenicity of neoantigen for combination immunotherapy of colorectal cancer" Science Advances (2020); Hyunjoon, K. et al. "Polymeric nanoparticles encapsulating novel TLR7/8 agonists as immunostimulatory adjuvants for enhanced cancer immunotherapy" Biomaterials (2018).

Concerns:

--The authors refer to "...antitumor immunity in colorectal adenocarcinoma" and conduct their preclinical in the MC38 model which is mouse cancer line of colorectal cancer (CRC) origin. However, other than the title no further refer is made within the manuscript to the relevance of their preclinical data to CRC. Referring to CRC in the title one would have expected clear mention to CRC including: the immuno-biology of the disease; potential relevance of the test agent; clinical un-met need; and translatability of preclinical findings.

--The authors use a single cancer line, MC38, in their in vitro studies and in vivo proof of concept work. No details were provided as to the rationale for selecting a single cancer cell line and or how many repeat and confirmatory in vivo studies were conducted. With such a paucity of model systems used in this study, it difficult to understand the credibility and relevance of PNP-R848 data and potential translatability to the clinic. Further, Fig 4. shows in vivo antitumor effects in the MC38 model; however the experiment lacks appropriate controls including PEG-NP and PNP without R848. The currently study is superficial with respect to data content and would be greatly enhanced if the in vitro characterization was conducted in a panel comprising of murine and human cancer cell lines. Moreover, if multiple murine syngeneic cancer models (e.g. n=3-4) were evaluated in vivo.

--The authors present a time-course in Fig. 2., comparing retention times of fluorescently labelled PEG-NP and PNP at the site of intratumoral injection. This study would be greatly enhanced if the two fluorescently labelled NPs had been loaded with R848 and supporting R848 release and pharmacokinetics (PK) data provided. If encapsulation of R848 is no feasible in fluorescently labelled NPs, then a supporting PK time-course should be provided in non-fluorescently labelled NPs. PK analysis of R848 (both at the IT site of injection and in blood compartments) is vital data to interpret the MC38 antitumor data shown in Fig. 4. Whilst PNP-R848 appears more efficacious of PEG-NP-R848, it is not clear whether the additive antitumor effects are a result of R848 priming/activating immune cells within the tumor microenvironment and or through R848 entering systemic circulation and targeting immune cells peripherally, within the blood and secondary organs of the immune system. This is a critical consideration when trying to understand the therapeutic margin of PNP-R848 and its potential clinical utility. This is particularly salient when one considers the in vitro BMDC release data show in Fig 3a., which depicts a clear dose-response of IL-6. Elevated systemic levels of IL-6 have been associated with cytokine release syndrome and poor tolerability in immunotherapy clinical studies.

--The authors present a range of in vitro in Fig. 3 incl. a dose response in a TLR7 reporter line (Fig 3a). Knowing R848 is a mixed TLR7 and 7 agonist, why is no data supplied with a TLR8 reporter line?

--To understand and address the therapeutic margin considerations of PNP-R848, as discussed previously (above), the authors need to show a wider range of cytokines in the BMDC release assay (Fig. 3d).

Reviewer #3 (Remarks to the Author): with expertise in Nanoparticles/drug delivery/TLR

Reviewer Comments

In this report, the authors reported the local delivery of the TLR agonist, resiquimod (R848), via platelet membrane-coated PLA nanoparticles (PNP-R848) to elicit potent antitumor responses in a colorectal tumor model. The platelet membrane coating provides a facile means of enhancing interactions with the tumor microenvironment, thereby maximizing the biological activity of R848 at low drug dosages. The results indicated that the intratumoral administration of PNP-R848 strongly enhances local immune activation and leads to complete tumor regression in 100% of mice, while providing absolute protection against repeated and aggressive tumor re-challenges. Overall, this study is innovative and has many interesting findings. The reviewer would like to recommend its acceptance in Nature Communications after reasonable revisions. The following are some questions need to be addressed before acceptance:

Questions and comments:

- 1, Stability tests of the NPs are suggested to be provided
2. Scale bar is suggested for Fig 1g
3. Fig 2a and 2c, why blank group and PEG-NP group are very similar
4. Figure 4b, why Free R848 group and PEG-NP-R848 are very similar?
5. What's reason for choosing PLA as NP material? And what's the molecular weight of the used PLA?
6. TEM image of uncoated PLA NPs is suggested to be provided
7. Did the authors compare the difference of local delivery and systemic delivery?

**Intratumoral immunotherapy using platelet-cloaked nanoparticles enhances
antitumor immunity in solid tumors**

Manuscript ID #: NCOMMS-20-20942

We are grateful to the Reviewers for their constructive comments and believe our paper has been greatly improved by their input. We agree with all of the Reviewers' suggestions and have responded fully to each point in the revised manuscript (changes are highlighted in blue). Our revisions are described below in a point-by-point manner.

Response Reviewer #1:

In this manuscript, Bahmani et al. are attempting to exploit a new immunotherapeutic strategy against cancer by engaging TLRs in the tumor micro-environment and inhibit tumor-promoting immune signalling. They use resiquimod (R848) small molecule immunomodulator, as a model of TLR7/8 agonists suggested to induce an effective antitumor immune response. The systemic administration of R848 (or other TLR7 agonists) exerts toxic effect that limit clinical translation, leading to the exploration of intratumoral injection. As direct intratumoral injection of R848 may lead to drug leakage in the blood circulation, the authors evaluated the possibility of use platelet membrane-cloaked nanoparticle (PNP) for intratumoral delivery of R848, based on the scientific rationale that the numerous markers of the platelet membranes would be expected to lead to enhanced adherence and retention within the tumor site, thereby resulting in stronger anti-cancer efficacy upon intratumoral administration.

To support their hypothesis the authors used a murine colorectal adenocarcinoma model and present data indicating that after intratumoral administration of the human platelet membrane coated PNP, there is an activation of APCs within the draining lymph node (DLN) and an increased immune infiltration. Further, they observed an efficient antitumoral response leading to short-term rejection of the induced tumors as well as long-term immunity tumor induction re-challenges.

We deeply appreciate the Reviewer's careful reading of our manuscript and the helpful comments provided. We have modified the manuscript accordingly.

Comments

The authors are asked to consider the following remarks:

Platelet membrane preparation and characterization: a more complete characterization of the platelet membrane for instance by DLS, TEM or SEM would be useful. The way used to prepare the membrane can very well lead to the formation of platelet extracellular vesicles which could "contaminate" the pellet during ultracentrifugation at 21' 100 x g (even for 10 min only). Do the authors truly obtain isolated platelet membranes or was there "contamination" with extracellular vesicles and potential experimental interference for the preparation of the PNP? Please comment.

We thank the Reviewer for this comment. We have performed detailed characterization of the platelet membrane in the past (*Nature* 2015, 526, 118) and thus decided to forgo it in the present work. While it is

possible that extracellular vesicles (EVs) may be present in our purified membrane preparation, they are unlikely to affect the overall function of the PNP nanoparticles as long as their surface composition is similar to the original platelets. Like with the platelets, the intravesicular components of the EVs would be emptied during our freeze-thaw process, and this is corroborated by the data from **Fig. 1b,c** showing little remaining thrombin and ADP in the purified membrane. During the coating process, the EV membrane would assemble around the nanoparticle cores in much the same way as platelet membrane, making them indistinguishable from each other. In the revised text, we have cited our previous work (*Nature* 2015, 526, 118) in case readers are interested in the detailed membrane characterization (see **Page 6**).

“Other detailed characterization of the platelet membrane has been previously reported³⁷.”

There is no data about drug loading efficacy of R848 in the NPs and little details about the loading methodology used which does not allow to reproduce the data. There are ample technical details about the (relatively standard) HPLC methodology to quantify R848...but by contrast essentially no details about what makes this engineering approach novel such as the coating methodology by sonication of the platelet membrane on the loaded or unloaded NP. Citing ref 37 (in the results section) may not be sufficient to be able to reproduce the technique. What are the conditions used for sonication? In addition, does the sonication affect the R-848 loading? What a polymer to membrane mass ratio of 1:0.7 means in non-equivocal terms? Does “the platelet membrane mass” means the protein content?

We thank the Reviewer for the suggestions. In the revised text, we have indicated the drug loading yield (see **Page 6**). Note that the loading quantification was performed after membrane coating, so it takes into account any effect of the sonication procedure.

“Quantification of drug loading revealed that 3.4 wt% of R848 could be encapsulated into the final formulation.”

Regarding the loading methodology, the information can be found in the methods section titled “Nanoparticle synthesis and physicochemical characterization” (see **Page 15-16**), and we have highlighted the relevant passage in the revised manuscript. We have also provided more information about the sonication procedure and clarified the membrane coating ratio (see **Page 16**).

“First, polylactic acid (PLA; R202H; Evonik) and R848 (BOC Sciences) were dissolved in an organic phase consisting of benzyl alcohol (Acros) and ethyl acetate (Fisher Chemical) at concentrations of 60 mg mL⁻¹ and 10 mg mL⁻¹, respectively. The mixture was then added to 5× volume of ice-cold outer phase media consisting of 10 mM Tris pH 7.5 (Invitrogen) with 0.2 wt% sodium cholate (Alfa Aesar) and 7 vol% ethyl acetate. This solution was homogenized at 12,000 rpm for 90 s using a Kinematica Polytron PT 3100 homogenizer before being passed through a Microfluidics LM20 Microfluidizer (outfitted with a Y chamber) three times. This mixture was then added to an equal volume of outer phase media, and the solvent was evaporated overnight in a fume hood while stirring at 200 rpm.”

“Platelet membrane coating was performed by sonication of the R848-loaded or unloaded nanoparticle cores with platelet membrane at a polymer to platelet membrane protein mass ratio of 1 to 0.7 for 2 min in a Branson CPX3800H ultrasonic bath at a frequency of 40 kHz.”

Why was the drug release kinetics of PNP-R848 was performed in the presence of 0.05% Triton X-100? I would strongly assume that this non-ionic detergent can artificially affect platelet membrane lipids, increase the permeability of the membrane and thus the release of R-848. This in itself could explain why the release “for both the bare NP-R848 and coated PNP-R848 formulations matched closely”. Please clarify why Triton X-100 is used and demonstrate that it does not affect the release from PMP-R848.

We thank the Reviewer for bringing up this point. In the original study, Triton X-100 was used at 0.05% as a mild detergent to facilitate release of R848 from the PNP-R848 formulation. To address the concern that this experimental setup may affect the membrane coating integrity, we have repeated the drug release kinetics without Triton X-100 and have updated the results in **Fig. 1h**. The new results suggest a similar pattern of R848 release from both bare NP-R848 and PNP-R848 at early timepoints (up to 8 hours). After this point, the kinetics significantly slowed, and we observed sustained release of R848 up to 6 days. The new data (see **Page 31**), discussion (see **Page 6**), and methods (see **Page 17-18**) have been added to the revised manuscript.

“Finally, the release of the R848 payload was studied over time, and the profiles for both the bare NP-R848 and coated PNP-R848 formulations matched closely, where more than 60% of the encapsulated payload was released within the first 24 h (Fig. 1h).”

Fig. 1. Nanoparticle characterization. h, Drug release profile from uncoated NP-R848 and coated PNP-R848 over 6 days.

Also it is much unclear how the 20 kDa dialysis cassettes was used. It seems to be an unusual manner to perform drug release experiments. Does it imply that the material was recirculated through the dialysis cassettes, as done for TFF? This may cause shear stress, speed-up the release, not mimicking well physiological situations. Please explain better how this is done and explain how this dialysis methodology was validated in your hands.

We apologize for the confusion regarding the drug release experiment. In brief, the sample is added to the dialysis cassettes, which are then placed into a large volume of PBS to create a sink for the drug to diffuse into. The large volume of solution in the sink is subjected to gentle stirring, while the sample remains in the dialysis cassette and is not circulated. With this setup, any released drug is able to diffuse out of the cassette into the larger sink volume. This is a gentle process that is commonly used to measure drug release in nanomedicine research (e.g. *J Ind Eng Chem* 2016, 24, 284; *RSC Adv* 2017, 7, 42073; *Drug Des*

Dev Ther 2015, 9, 2301). In the revised text, we have further clarified the details of the drug release study (see **Page 17-18**).

“Drug release kinetics from PNP-R848 were evaluated utilizing 20 kDa dialysis cassettes (Thermo Scientific). Reconstituted samples were transferred to the cassettes via a syringe with a 21-gauge needle and dialyzed against a large volume of PBS. The dissolution experiments were run at room temperature while stirring the PBS solution at 280 rpm for 6 days. Samples were collected at various timepoints (30 min, 1 h, 2 h, 4 h, 6 h, 8 h, 1 day, 2 days, 3 days, and 6 days) and analyzed by UHPLC.”

How were the various NPs sterilized for in vitro and in vivo experiments?

This information has also been added to the revised methods (see **Page 16**).

“All formulations were passed through a 0.2- μ m filter prior to use in vitro or in vivo experiments.”

MC38 binding and uptake: the incubation was performed at 4 °C for determination of binding. Can the use of 4°C artificially affect the cells and impact the binding data?

The assessment of nanoparticle binding to the cell surface at 4 °C is a standard procedure and is widely employed in the field of nanomedicine (e.g. *Lasers Surg Med* 2014, 46, 582; *Biochim Biophys Acta Biomembr* 2006, 1758, 713; *Mol Pharm* 2015, 11, 2989). By performing the experiment at 4 °C, active cellular processes such as endocytosis are suppressed, leaving only the physical binding interactions to account for increases in nanoparticle signal on the cells. This point has been further clarified in the revised text (see **Page 18**).

“The incubation was performed for 30 min at 4 °C in order to minimize endocytic uptake, after which the cells were washed three times with PBS and examined using flow cytometry.”

A major issue that may invalidate the claimed data is the fact that the authors used human platelets to prepare the membrane coated NP and used C57BL/6 immunocompetent mice for the various explorations, including interaction with immune cells. To which extent are the noted impact on e.g. BMDC, activation of APCs within the DLN and, most critically, increased immune infiltration linked to the functional design of the NP, or simply to an inter-species human-mice immune reactivity (which would then make the system inefficient in humans using human platelets coated-NP-R848)? Also I did not see whether the BMDC are human or rodent cells. Was a validation showing similar effects done using NP coated with platelet membranes from mice performed by the author?

The Reviewer brings up a valid point, as we elected to use human platelets to develop our PNP formulations, which were then evaluated using either murine BMDCs in vitro or murine tumor models in vivo. The main reason for this choice was for clinical translation considerations, as discussions with the FDA have revealed that they prefer for us to develop and evaluate the final drug formulation that will be used in humans, even when conducting the nonclinical studies in animal models. In our case, it is the human PNP-R848 formulation. To alleviate concerns of potential cross-species immune reactivity driving the therapeutic efficacy, we have conducted additional experiments using murine BMDCs, studying

whether there is a difference between empty PNPs fabricated using mouse or human platelet membrane. In **Fig. S2**, we can clearly see that neither formulation leads to the upregulation of CD80 or CD86, suggesting that the human membrane does not induce mouse BMDC maturation. Further, when looking at cytokine secretion, the empty platelet membrane-coated nanoparticles again had little effect. In contrast, human PNPs loaded with R848 induced a significant amount of cytokine secretion, confirming that the majority of the biological effect of our formulation results from delivery of the drug. The new results (see **Page S6**), along with the associated discussion (see **Page 8**) and methods (see **Page S2**), have been added to the revised manuscript.

“It was also confirmed that empty PNP, regardless of whether the platelet membrane was sourced from humans or mice, did not induce appreciable APC maturation or cytokine secretion (Supplementary Fig. 2).”

Figure S2. In vitro activity of empty PNP. a,b, Expression of CD80 (a) and CD86 (b) by BMDCs after incubation with empty PNP fabricated with human platelet membrane (hPNP) or mouse platelet membrane (mPNP). c-e, Secretion of IL-6 (c), TNFα (d), and IL-12p40 (e) by BMDCs after incubation with hPNP, mPNP, free R848, hPNP-R848, or LPS (mean ± SD).

“Nanoparticle activity on BMDCs

For co-stimulatory marker and cytokine release characterization, BMDCs from C57BL/6 mice were plated at 5×10^5 cells per well in 6-well plates. Then, empty PNP fabricated from mouse platelets (mPNP) or human platelets (hPNP), free R848, or PNP-R848 were added to the cells at an R848 concentration of 24.6 ng mL⁻¹ (or equivalent nanoparticle amount) and incubated at 37 °C for 24 h. Lipopolysaccharide (LPS; Invivogen) at 20 ng mL⁻¹ was used as a positive control. Afterwards, the supernatant was assayed using ELISA kits for IL-6, IL-12p40, and TNFα. The cells were washed and scraped from the plates, followed by staining with FITC-conjugated anti-mouse CD45, PE-conjugated

anti-mouse CD80, and APC-conjugated anti-mouse CD86. Data was collected using a Becton Dickinson Accuri C6 flow cytometer and analyzed with Flowjo software.”

The statistical methods are not described. Please provide the missing information.

Where appropriate, all of the statistical tests (i.e. one-way ANOVA, log-rank test) that were performed and the definitions for significance can be found in the respective caption for the corresponding figure (see **Page 32, 35, S8**).

*“Fig. 2. Nanoparticle interaction with tumor cells. a,b, Quantification of binding (a) and uptake (b) of PEG-NP and PNP by various cancer cells (MC38, HT-29, 4T1, and MDA-MB-231) after incubation in vitro (mean + SD; MFI = mean fluorescence intensity). * $p < 0.05$, ** $p < 0.01$, *** $p < 0.001$, **** $p < 0.0001$ (compared with PNP); two-way ANOVA.”*

*“Fig. 5. Immune response to treatment in tumor-bearing mice. f, Representative histological sections from the experiment in (e) (scale bar = 100 μm ; brown = positive staining). * $p < 0.05$, ** $p < 0.01$, *** $p < 0.001$, **** $p < 0.0001$ (compared with PNP-R848); one-way ANOVA.”*

*“Fig. 6. Therapeutic antitumor efficacy in a 4T1 tumor model. g, Number of metastatic nodules in the lungs at day 30 post-treatment (mean + SD). * $p < 0.05$, *** $p < 0.001$, **** $p < 0.0001$ (compared with PNP-R848); one-way ANOVA.”*

The presence of phosphatidylserine a potent pro-coagulant phospholipid on the platelet membranes seems to contradict the fact that assays for thrombin and adenosine diphosphate did not reveal a prothrombotic activity. How can this be explained? In addition, the precise experimental conditions used to carry these assays are not described in the method section. It should be, as well as the way how control materials (PRP, platelet lysate) were prepared. The evil is in the details. Please add experimental details e.g. as supplementary file.

We thank the Reviewer for bringing up these points. It should be noted that the main goal of the thrombin and ADP assays were to confirm removal of the platelets' intracellular contents in our membrane derivation process. The absence of the two components in the final membrane preparation confirms their physical removal, but it cannot necessarily be used to indicate the activation state of the membrane or the original platelets. Due to the sensitive nature of platelets, we fully acknowledge that activation can occur during handling. Fortunately, the presence of some phosphatidylserine on the nanoformulation is unlikely to negatively impact immunotherapeutic efficacy or safety, particularly since we are studying the intratumoral route of administration. We have attempted to further clarify the purpose of the assays in the revised text (see **Page 6**). Additionally, we have provided methods to describe the quantification assays from thrombin and ADP (see **Page 16-17**).

“Quantification of platelet activation factors. PRP, platelet lysate, and purified platelet membrane were prepared and examined for the platelet-activating molecules thrombin and adenosine diphosphate (ADP) using a Sensolyte 520 thrombin activity assay kit (Anaspec) and a PicoProbe ADP assay kit (BioVision), respectively, based on the manufacturers' instructions. To prepare the PRP, lysate, and platelet

membrane samples, 1 mL of PRP was first diluted 3× and divided into 3 groups. The first group was used directly, the second group was washed and underwent one freeze-thaw cycle to produce the platelet lysate, and the third group was processed to completion into purified membrane. All samples were resuspended to the same volume with PBS prior to performing the assays.”

The authors seem to administer a dose of 30 μL into 30 - 40 mm^3 tumors. This is a very high dose for what are actually quite small tumors. Please comment in the discussion.

It is true that 30 μL is a significant volume to directly administer into small tumors. While it was possible for us to administer the same dosage of nanoparticles in a smaller volume, we decided against it due to the increased variability that injecting small volumes by syringe would introduce. In order to accommodate the larger volume, we administered the solution slowly to prevent pressure build up that would cause the solution to leak out of the tumor. We have clarified this point in the revised text (see **Page 21**).

“Care was taken to administer the solution slowly to prevent leakage.”

A progression-free survival defined as tumor volume $< 200 \text{ mm}^3$ seems to be a small value. Please clarify.

This is a good point brought up by the Reviewer. In our case, we set the threshold for progression-free survival at 200 mm^3 based on our experience with the tumor models that were studied in this work. Due to the relatively aggressive nature of these syngeneic tumors, those growing larger than 200 mm^3 are rarely rejected and continue to grow, even after aggressive immunotherapy. In contrast, we observed that many tumors under this threshold could reverse course and respond completely to treatment. We have clarified our selection of the threshold for progression-free survival in the revised manuscript (see **Page 21**).

“Tumor growth and mouse weight were monitored every other day. Progression-free survival was defined as tumor volume $< 200 \text{ mm}^3$, a threshold above which complete tumor regression is unlikely.”

Fig 1. G: as early kinetics of release is more important, a zoom on the 0-10 hours earlier period could be added (minor comment)

To address the Reviewer’s comment, we have added an inset into the new **Fig. 1h** in order to facilitate visualization of the earlier timepoints (see **Page 31**).

Fig. 1. Nanoparticle characterization. h, Drug release profile from uncoated NP-R848 and coated PNP-R848 over 6 days.

Fig 1.F Only 2 nanoparticles are shown. This does not give an idea of a whole population (e.g. 50-100) of NPs, in particular the risk of aggregation. In addition, comparative cryo-TEM pictures of the bare NP and PNP should be presented.

We agree with the Reviewer and have performed additional TEM imaging on the bare NP-R848 formulation, as well as the PNP-R848 formulation. In **Fig. 1f,g**, we can visualize the difference between coated and uncoated nanoparticles. In **Fig. S1a**, we have presented multiple lower magnification images of the PNP-R848 formulation. The new data (see **Page 31, S5**) have been included in the revised manuscript along with the appropriate discussion (see **Page 6**) and additional methods (see **Page S2**).

“Transmission electron microscopy revealed that, compared to bare NP-R848, the final PNP-R848 formulation possessed a core-shell structure with a layer of membrane coating on the outside (Fig. 1f,g and Supplementary Fig. 1a).”

Fig. 1. Nanoparticle characterization. f,g, Transmission electron microscopy visualization of uncoated NP-R848 (f) and coated PNP-R848 (g) with uranyl acetate negative staining (scale bars = 50 nm).

Figure S1. Morphology and stability characterization of PNP-R848. a, Transmission electron microscopy visualization (multiple fields of view) of PNP-R848 with uranyl acetate negative staining (scale bars = 200 nm).

Fig 1.D The increase in size after coating is around 25 nm. Which brings the platelet membrane thickness to 12.5 nm. Does this corresponds to the actual platelet membrane thickness?

This is an astute observation by the Reviewer. The actual physical thickness of the platelet membrane is less than the 12.5 nm value suggested by the dynamic light scattering (DLS) data. This is to be expected, as DLS measures the hydrodynamic size of particles in suspension, and this takes into account the electric dipole layer. The thickness of this layer depends on various factors, such as the electrical conductivity of the liquid. We have clarified this point in the revised text (see **Page 6**).

“Physicochemical characterizations revealed that membrane coating slightly increased the hydrodynamic size of both the bare PLA nanoparticle cores as well as the bare R848-loaded nanoparticle cores (NP-R848) (Fig. 1d).”

Response to Reviewer #2:

The authors of this manuscript designed an intratumoral (IT) platelet membrane-coated nanoparticle (PNP) to promote local localized delivery of the TLR agonist resiquimod (R848) with an aim to induce anti-tumour immunity effects whilst ameliorating systemic side-effects. They showed IT dosing of the PNP loaded with R848 (PNP-R848) induced tumor regression in a single syngeneic mouse cancer model, and animals with a complete response to PNP-R848 demonstrated immunity to tumor re-challenge. By flow cytometry and immunohistochemistry analysis of draining lymph node (DLN) and tumor tissues(s) they describe enhanced immune activation in the PNP-R848 treatment arm versus a “more traditional” R848 formulation. Not much of the information provided in this manuscript is really novel or unexpected. In addition to the preclinical oncology studies referenced in this manuscript, describing encapsulation of adjuvants, such as TLR agonists, within nanoparticles (e.g. Schmid, D. et al Nat. Comms. (2018); Da Silva, CG. et al. Biomaterials (2019) there is a surfeit of other papers on this topic e.g. Qianqian, N. et al. “A bi-adjuvant nanovaccine that potentiates immunogenicity of neoantigen for combination immunotherapy of colorectal cancer” Science Advances (2020); Hyunjoon, K. et al. “Polymeric nanoparticles encapsulating novel TLR7/8 agonists as immunostimulatory adjuvants for enhanced cancer immunotherapy” Biomaterials (2018).

We have found the Reviewer’s comments to be very helpful in improving our manuscript. All the comments are valid, and we have performed additional experiments, as well as provided additional discussion. We truly appreciate the time that the Reviewer has taken to provide such valuable feedback. Meanwhile, we respectfully argue that using platelet membrane-coated nanoparticles (PNPs) to deliver TLR agonist resiquimod (R848) for intratumoral immunotherapy is novel and holds high potential for clinical translation. Moreover, the reported PNP-R848 formulation represents a platform that can be applied for multiple solid tumors.

Concerns:

--The authors refer to “...antitumor immunity in colorectal adenocarcinoma” and conduct their preclinical in the MC38 model which is mouse cancer line of colorectal cancer (CRC) origin. However, other than the title no further refer is made within the manuscript to the relevance of their preclinical data to CRC. Referring to CRC in the title one would have expected clear mention to CRC including: the immuno-

biology of the disease; potential relevance of the test agent; clinical un-met need; and translatability of preclinical findings.

We thank the Reviewer for the comment. In light of the new data that we have obtained to demonstrate that our PNP-R848 platform can be generalized beyond colorectal cancer (please see our response to the next comment), we have updated the title to reflect this (see **Page 1**). Ultimately, we believe our approach for intratumoral immunotherapy can be successfully expanded to encompass many different types of solid tumors, and this is a point that we further emphasize in the introduction (see **Page 4**).

“Intratumoral immunotherapy using platelet-cloaked nanoparticles enhances antitumor immunity in solid tumors”

“Herein we report on the development of a platelet membrane-cloaked nanoparticle (PNP) specifically for the intratumoral delivery of R848 to treat solid tumors.”

--The authors use a single cancer line, MC38, in their in vitro studies and in vivo proof of concept work. No details were provided as to the rationale for selecting a single cancer cell line and or how many repeat and confirmatory in vivo studies were conducted. With such a paucity of model systems used in this study, it difficult to understand the credibility and relevance of PNP-R848 data and potential translatability to the clinic. Further, Fig 4. shows in vivo antitumor effects in the MC38 model; however the experiment lacks appropriate controls including PEG-NP and PNP without R848. The currently study is superficial with respect to data content and would be greatly enhanced if the in vitro characterization was conducted in a panel comprising of murine and human cancer cell lines. Moreover, if multiple murine syngeneic cancer models (e.g. n=3-4) were evaluated in vivo.

We thank the Reviewer for this comment and largely agree with the assessment. As such, we have performed more in vitro experiments and expanded testing to a panel of murine and human cancer cells. In **Fig. 2a,b**, we evaluated binding and uptake of the PNP formulation by a human colorectal cancer cell line (HT-29), mouse breast cancer cell line (4T1), and human breast cancer cell line (MDA-MB-231). Similar to the original MC38 data, we see that both binding and uptake are significantly enhanced with the platelet membrane coating. The new data (see **Page 32**) has been included with the relevant discussion (see **Page 6-7**) and methods (see **Page 18**) in the revised text.

“It was observed by flow cytometry that PNP much more readily bound to all four cancer cells compared with a polyethylene glycol (PEG)-coated nanoparticle (PEG-NP) control (Fig. 2a). These results correlated well with cellular uptake, which was also significantly higher for PNP than for PEG-NP in all of the cell lines (Fig. 2b).”

Fig. 2. Nanoparticle interaction with tumor cells. *a,b*, Quantification of binding (*a*) and uptake (*b*) of PEG-NP and PNP by various cancer cells (MC38, HT-29, 4T1, and MDA-MB-231) after incubation *in vitro* (mean + SD; MFI = mean fluorescence intensity). * $p < 0.05$, ** $p < 0.01$, *** $p < 0.001$, **** $p < 0.0001$ (compared with PNP); two-way ANOVA.

It should be noted that we studied the potential antitumor effects of PEG-NP and PNP without R848 in **Fig. S4a**. Here, we showed that neither of the nanoparticles had a significant impact on tumor growth. We can thus be fairly confident that the therapeutic effect observed in our antitumor efficacy data resulted from the enhanced delivery of R848 by our PNP-R848 formulation. For the MC38 model, we can confirm that the results we presented are representative of more than 3 independent experiments.

Figure S4. Therapeutic antitumor efficacy of empty nanocarriers. *a*, Progression-free survival (tumor size <math> < 200 \text{ mm}^3 </math>) of mice after treatment with PEG-NP or PNP without R848 loading (NS = not significant, log-rank test).

To further demonstrate that our platform has broad applicability, we studied antitumor efficacy in another syngeneic tumor model. In the new **Fig. 6**, it can be seen that treatment with the PNP-R848 formulation had a marked impact on the growth of 4T1 tumors. Additionally, we also looked at the ability of the treatment to reduce metastasis. Whereas control mice had on average more than 50 metastatic nodes per lung at day 30 post-treatment, the mice treated with PNP-R848 had almost none. We believe that this data successfully demonstrates the ability of PNP-R848 to induce strong systemic immunity upon intratumoral administration. The data (see **Page 36**) has been included, along with additional discussion (see **Page 5, 12, 14**) and updated methods (see **Page 22-23**), in the revised text.

“Antitumor efficacy in a mouse breast cancer model. To further evaluate the applicability of PNP-R848 as a generalized treatment against solid tumors, anticancer efficacy was tested in a syngeneic murine 4T1 triple-negative breast cancer model established using BALB/c mice (**Fig. 6a**). Each animal was

subcutaneously implanted with 5×10^5 tumor cells in the right flank, and the average tumor size was allowed to reach $\sim 30\text{-}40 \text{ mm}^3$ before treatment with either 8% sucrose, free R848, PEG-NP-R848, or PNP-R848 at a drug dosage of $15 \mu\text{g}$ per injection. The mice were treated every other day for a total of 5 times, and the tumor sizes and progression-free survival were monitored (Fig. 6b-d). Similar to the MC38 model, administration of PNP-R848 resulted in significant inhibition of 4T1 tumor growth. With PNP-R848 treatment, progression-free survival was prolonged to 23 days, compared to 9 days for the control group. Both free R848 and PEG-NP-R848 exhibited an intermediate level of antitumor efficacy. This trend was also reflected on day 30 after the first treatment, when the tumors were excised and weighed (Fig. 6e,f). Notably, PNP-R848 had a marked effect on the number of metastatic nodules in the lungs, reducing the average number per lung to 3 nodules from more than 50 for the control group (Fig. 6g).”

Fig. 6. Therapeutic antitumor efficacy in a 4T1 tumor model. *a*, Schematic timeline of the efficacy study. Tumors were treated with free R848, PEG-NP-R848, or PNP-R848. *b*, Average tumor growth kinetics after treatment (mean \pm SEM). *c*, Individual tumor growth kinetics after treatment. *d*, Progression-free survival (tumor size $< 200 \text{ mm}^3$) of mice after treatment. *e*, Images of tumors at day 30 post-treatment. *f*, Average tumor weights at day 30 post-treatment (mean \pm SD). *g*, Number of metastatic nodules in the

*lungs at day 30 post-treatment (mean + SD). *p < 0.05, ***p < 0.001, ****p < 0.0001 (compared with PNP-R848); one-way ANOVA.*

--The authors present a time-course in Fig. 2., comparing retention times of fluorescently labelled PEG-NP and PNP at the site of intratumoral injection. This study would be greatly enhanced if the two fluorescently labelled NPs had been loaded with R848 and supporting R848 release and pharmacokinetics (PK) data provided. If encapsulation of R848 is no feasible in fluorescently labelled NPs, then a supporting PK time-course should be provided in non-fluorescently labelled NPs. PK analysis of R848 (both at the IT site of injection and in blood compartments) is vital data to interpret the MC38 antitumor data shown in Fig. 4. Whilst PNP-R848 appears more efficacious cf PEG-NP-R848, it is not clear whether the additive antitumor effects are a result of R848 priming/activating immune cells within the tumor microenvironment and or through R848 entering systemic circulation and targeting immune cells peripherally, within the blood and secondary organs of the immune system. This is acritical consideration when trying to understand the therapeutic margin of PNP-R848 and its potential clinical utility. This is particularly salient when one considers the in vitro BMDC release data show in Fig 3a., which depicts a clear dose-response of IL-6. Elevated systemic levels of IL-6 have been associated with cytokine release syndrome and poor tolerability in immunotherapy clinical studies.

This is an excellent point that the Reviewer has brought up. We agree that analyzing the concentrations of R848 in vivo will help us to better understand the mechanism and therapeutic margins for our platform. To address the comment, we have performed additional pharmacokinetic studies to quantify tumor and plasma levels of R848 after intratumoral administration of our nanoformulation. In **Fig. 2e,f**, we can see that, similar to the fluorescently labeled nanoparticles, the drug is better retained at the tumor site for the PNP-R848 group compared with PEG-NP-R848. This increased drug retention over the course of several days likely explains the improved antitumor efficacy of the PNP-R848 formulation. Further, we see that both PEG-NP-R848 and PNP-R848 exhibit very limited systemic drug exposure. This is in contrast to free R848, which displays a large spike right after intratumoral injection. The data suggests that PNP-R848 is effective at staying within the tumor microenvironment, where it slowly releases the drug payload. On the other hand, PEG-NP-R848, despite not releasing drug into circulation, is likely draining away from the tumor site due to its lack of targeting functionality. The new data (see **Page 32**) has been included in the revised text, along with the relevant discussion (see **Page 7**) and methods (see **Page 20-21**).

“We then analyzed in vivo drug levels after intratumoral administration of R848-loaded nanoparticles. Compared with injection of free R848, which resulted in a large transient spike in serum R848 concentration, both PEG-NP-R848 and PNP-R848 were able to significantly limit systemic exposure (Fig. 2e). However, only PNP-R848, with its enhanced affinity to tumor cells, enabled prolonged drug persistence in the tumor tissue (Fig. 2f).”

Fig. 2. Nanoparticle interaction with tumor cells. e, Plasma levels of R848 after intratumoral administration of free R848, PEG-NP-R848, and PNP-R848 into mice bearing MC38 tumors (mean \pm SEM). **f,** Retention of R848 after intratumoral administration of PEG-NP-R848 and PNP-R848 into mice bearing MC38 tumors (mean \pm SEM).

--The authors present a range of in vitro in Fig. 3 incl. a dose response in a TLR7 reporter line (Fig 3a). Knowing R848 is a mixed TLR7 and 7 agonist, why is no data supplied with a TLR8 reporter line?

We thank the Reviewer for this helpful comment and have assessed the dose-dependent response to PNP-R848 using a TLR8 reporter line. As seen in the new **Fig. 3b**, there is a strong response against both free R848 and PNP-R848, whereas empty PNP does not display significant activity. The new data (see **Page 33**), along with the relevant discussion (see **Page 7-8**) and methods (see **Page 18**), have been added to the revised manuscript.

“To directly assess the biological activity of the R848 payload, we incubated PNP-R848 with human reporter cell lines expressing either TLR7 or TLR8, which provide a colorimetric readout in response to NF- κ B activation (Fig. 3a,b). The cells were incubated with free R848 or PNP-R848 for 21 h, and our results showed the activities of the two were roughly equivalent at the same drug concentration. As expected, PNP nanoparticles without drug loading showed minimal TLR7 and TLR8 activation.”

Fig. 3. Nanoparticle in vitro activity and interaction with immune cells. a,b, Dose-dependent response of TLR7 (a) and TLR8 (b) reporter cell lines after incubation with PNP, free R848, and PNP-R848 (mean \pm SD).

--To understand and address the therapeutic margin considerations of PNP-R848, as discussed previously (above), the authors need to show a wider range of cytokines in the BMDC release assay (Fig. 3d).

This is a good point brought up by the Reviewer. We have further measured the release of TNF α and IL-12 after treating mouse BMDCs with PNP-R848, and the results are presented in the new **Fig. 3e-g**. From the new data, we can see that the PNP-R848 largely retains the activity of the R848 when incubated with BMDCs in vitro. There is a dose-dependent increase in cytokine production that is very similar to the trend that we observed for IL-6. The new plots (see **Page 33**), relevant discussion (see **Page 8**), and methods (see **Page 19**) have been included in the revised text.

“Additionally, we assessed the ability of PNP-R848 to elicit the production of proinflammatory cytokines such as IL-6, tumor necrosis factor α (TNF α), and IL-12 by BMDCs (Fig. 3e-g). After incubation with various concentrations of free R848 or PNP-R848, the culture supernatant was analyzed by enzyme-linked immunosorbent assays (ELISAs). For each cytokine that was studied, our results showed a dose-dependent release pattern that was similar for both samples.”

Fig. 3. Nanoparticle in vitro activity and interaction with immune cells. e-g, Dose-dependent secretion of IL-6 (e), TNF α (f), and IL-12p40 (g) by BMDCs after incubation with free R848 and PNP-R848 (mean \pm SD).

Response to Reviewer #3

Reviewer Comments

In this report, the authors reported the local delivery of the TLR agonist, resiquimod (R848), via platelet membrane-coated PLA nanoparticles (PNP-R848) to elicit potent antitumor responses in a colorectal tumor model. The platelet membrane coating provides a facile means of enhancing interactions with the tumor microenvironment, thereby maximizing the biological activity of R848 at low drug dosages. The results indicated that the intratumoral administration of PNP-R848 strongly enhances local immune activation and leads to complete tumor regression in 100% of mice, while providing absolute protection against repeated and aggressive tumor re-challenges. Overall, this study is innovative and has many interesting findings. The reviewer would like to recommend its acceptance in Nature Communications after reasonable revisions. The following are some questions need to be addressed before acceptance:

We are grateful to the Reviewer for kind assessment of our work and providing thoughtful comments. We have addressed all the comments in the revised manuscript.

1, Stability tests of the NPs are suggested to be provided.

This is a good suggestion by the Reviewer, and we have assessed the stability of the NP-R848 and PNP-R848 formulations in PBS over the course of 4 weeks, both at room temperature and at 4 °C. In the new **Fig. S1b**, we can see that there is little change in size over this period. The new data (see **Page S5**), the relevant discussion (see **Page 6**), and the methods (see **Page S2**) have been added to the revised text.

“The nanoparticles remained stable in PBS over the course of 4 weeks, both when stored at room temperature and at 4 °C (Supplementary Fig. 1b).”

Figure S1. Morphology and stability characterization of PNP-R848. b, Size of NP-R848 and PNP-R848 in PBS over 4 weeks at room temperature (RT) or at 4 °C (4C) (mean ± SD).

2. Scale bar is suggested for Fig 1g

The scale bar information for **Fig. 1f,g** are now included in the caption (see **Page 31**).

“Fig. 1. Nanoparticle characterization. f,g, Transmission electron microscopy visualization of uncoated NP-R848 (f) and coated PNP-R848 (g) with uranyl acetate negative staining (scale bars = 50 nm).”

3. Fig 2a and 2c, why blank group and PEG-NP group are very similar.

This is a good observation by the Reviewer. Due to the stealth properties of its PEG coating, PEG-NP is expected to bind to cancer cells significantly less compared with PNP. However, we acknowledge that a certain level of PEG-NP uptake should be expected. In the original study, the experimental setup resulted in the difference between PEG-NP and the control group being indiscernible. We have repeated the study, shown in the new **Fig. 2a,b**, with multiple cell lines and have shown that the correct trend is now observed. The new data (see **Page 32**), along with the appropriate discussion (see **Page 6-7**) and methods (see **Page 18**), have been included in the revised text.

“It was observed by flow cytometry that PNP much more readily bound to all four cancer cells compared with a polyethylene glycol (PEG)-coated nanoparticle (PEG-NP) control (Fig. 2a). These results correlated well with cellular uptake, which was also significantly higher for PNP than for PEG-NP in all of the cell lines (Fig. 2b).”

4. Figure 4b, why Free R848 group and PEG-NP-R848 are very similar?

We believe that the free R848 and PEG-NP-R848 resulted in similar antitumor efficacy due to lower tumor retention. The lower tumor drug retention for PEG-NP-R848 compared with our PNP-R848 formulation was confirmed in a new pharmacokinetics data in **Fig. 2e,f**. While PEG-NP-R848 does not result in as much systemic exposure as free R848, drainage away from the tumor site combined with delayed drug release likely accounts for its reduced efficacy. The new data (see **Page 32**) has been included in the revised manuscript along with the appropriate discussion (see **Page 7**) and methods (see **Page 20-21**).

“However, only PNP-R848, with its enhanced affinity to tumor cells, enabled prolonged drug persistence in the tumor tissue (Fig. 2f). Taken together, these studies demonstrated that the platelet membrane coating, which displays surface markers known to play a role in cancer cell binding⁵¹, was able to significantly increase nanoparticle affinity to the MC38 tumor cells compared with a more traditional PEG coating.”

5. What’s reason for choosing PLA as NP material? And what’s the molecular weight of the used PLA?

PLA was chosen as the nanoparticle core material because it has been established as a biocompatible and biodegradable polymer that is classified as Generally Recognized as Safe (GRAS) by the FDA. It is used in many resorbable surgical devices such as sutures, ligatures, and meshes. The PLA used in these studies (Resomer 202H) is approximately 10-18 kDa in molecular weight. We have further clarified the selection of PLA in the revised introduction (see **Page 5**).

“This was done by directly coating the membrane isolated from human platelets through a differential centrifugation and freeze-thaw process onto biocompatible and biodegradable polylactic acid (PLA) nanoparticle cores via sonication³⁷.”

6. TEM image of uncoated PLA NPs is suggested to be provided.

This is a good suggestion by the Reviewer. We have added an additional TEM image for the uncoated NP-R848 sample as **Fig. 1f** (see **Page 31**), along with the relevant discussion (see **Page 6**).

“Transmission electron microscopy revealed that, compared to bare NP-R848, the final PNP-R848 formulation possessed a core-shell structure with a layer of membrane coating on the outside (Fig. 1f,g and Supplementary Fig. 1a).”

Fig. 1. Nanoparticle characterization. f,g, Transmission electron microscopy visualization of uncoated NP-R848 (f) and coated PNP-R848 (g) with uranyl acetate negative staining (scale bars = 50 nm).

7. Did the authors compare the difference of local delivery and systemic delivery?

We thank the Reviewer for this comment. It should be noted that the focus of this study was specifically to evaluate the PNP platform for intratumoral delivery of R848 as an immunotherapy. As discussed in our introduction, there are certain drawbacks to systemically delivered therapies that we wished to overcome using the PNP formulation, including safety concerns and transient local immune insufficiency. To avoid any confusion, we have further emphasized the goal of our study in the revised introduction (see **Page 4**).

“Herein we report on the development of a platelet membrane-cloaked nanoparticle (PNP) specifically for the intratumoral delivery of R848 to treat solid tumors. The plasma membrane derived from human platelets, with its multitude of proteins, glycoproteins, and lipids, bestows platelet-mimicking properties such as selective adherence to cells in the tumor microenvironment³⁷.”

REVIEWER COMMENTS

Reviewer #1 (Remarks to the Author):

I thank Bahmani et al. for responding to the issues I raised on the initial submission. The following substantial concerns remain:

a- Platelet membrane preparation and characterization: referring to a previous publication is not sufficient in my opinion to address the concerns I have been raising regarding the potential presence of EVs. In addition, it should not be taken for granted that the EVs would be, like platelets, emptied by a freeze-thaw cycle. Actually, freezing is one standard way used to store isolated EVs and studies have shown that the vesicular structure, size and count of EV remain unaffected after freeze-thaw. In addition, even if the EVs are indeed disrupted by the freeze-thaw cycle, assuming that their membranes would readily assemble around the nanoparticles in much the same way as platelet membranes is very speculative.

b- Drug loading efficacy: I thank the authors for providing more detailed experimental information. However, the authors claim a 3.4wt% of R848 encapsulated into the final formulation. How did the authors verify that this amount is truly encapsulated within the NP and not a residual amount co-purified with the nanoformulation?

c- I thank the reviewers for performing the additional experiment of drug release in the absence of 0.05% Triton X-100. The fact that the kinetics of release of R848 from the PMP-R848 is very similar with and without Triton X-100 now questions (a) the need for its addition in the first place and (b) whether the PMP-R848 are truly coated, or maybe imperfectly coated, with platelet membranes... which is worrisome

d- Dialysis method: the authors have answered my concerns in a clear manner

e- I thank the reviewers for documenting that the sterilization of the NPs was achieved by 0.2-micrometre filtration. Was there any loss of NPs at this stage and was it validated that this step does not "uncoat" the PMP-R848?

f- Use of human platelets: I understand the argument provided by the authors that the selection of human platelet membrane is logical having in mind considerations for clinical translation. I am also aware that regulatory agencies prefer that pre-clinical studies done within the scope of an IND or registration for a human-tissue product be carried out with the human product itself, especially to assess safety aspects. Such pre-clinical studies are carried out having in mind the risks of immunological reactions from the animal used for such studies. I am doubtful, though, that regulators would be much convinced by the studies carried out in this manuscript... Still, I commend the additional in vitro experiments done by the authors but I remain sceptical that they could reflect what happens in vivo. In addition in Fig S2a & b, why are the peaks obtained in flow-cytometry, especially the blank, so broad? Does it reflect a baseline "upregulation"? In Fig S2C, d, & e, showing also mPNP-848 as control would be justified.

g- Statistical analyses: the authors have addressed my concerns.

h- Phosphatidylserine expression and pro-coagulation risks: the authors have addressed my concerns.

i- Dose of 30 μ L into 30 - 40 mm³ tumours: the authors provided some words of caution (but it remains a large volume for this tumour size).

j- Tumor volume < 200 mm³: the authors should provide a reference with their answer, or indicate that the value was selected based on their own experimental experience.

k- Fig 1. G: early kinetics of release has been added

l- Fig 1.F: The authors provide additional TEM observation. It is quite unclear to my eyes what are the morphological differences between the uncoated and platelet membrane coated nanoparticles. The authors should precisely indicate in figure g where one can observe the platelet membrane. In Fig S1a I am a bit puzzled by the variation in size of the PNP-R848 populations shown. What is actually the variation in size seen in the uncoated starting NPs? Could the apparent heterogeneity in the PNP-R848 population due to the presence of platelet EVs, as discussed above?

m- Fig 1.D: Increase in size: the response from the authors is accepted but remains speculative.

Reviewer #2 (Remarks to the Author):

Reviewer #2 response:

(1) Happy with author(s) change to manuscript title.

(2) Happy with author(s) new (murine and human) cell line panel incl. text and Figure 2.

(3) Not sure I am convinced by data presented in Figure S4 since there appears to be an antitumor effect with PNP without R848 i.e. 25% of mice have tumor < 200 mm³ at 30-days; whilst 100% of the two control groups tumors (Control; PEG-NP) have grown > 200 mm³ at 30-days.

(4) Happy with author(s) new antitumor efficacy data in 4T1 model highlighted in text and Figure 6.

(5) New PNP-R848 PK data (Fig. 2e,f,..) is welcome. However hard to interpret relevance of result since no information provided on PNP-R848 dose used in Fig. 2e,f,. Moreover, how this dose relates to the PNP-R848 dose used in the MC38 efficacy study (Fig. 4.). This dataset would be further improved with respect to safety risk if systemic cytokines had been included.

(6) Happy with author(s) additional TLR8 reporter data incl. text Fig. 3.

(7) Happy with author(s) additional cytokine data incl. text Fig. 3.

Reviewer #3 (Remarks to the Author):

The revised manuscript has addressed all the comments and questions raised by the reviewers and I am satisfied with the answers. Therefore, it's OK for acceptance now.

**Intratumoral immunotherapy using platelet-cloaked nanoparticles enhances
antitumor immunity in solid tumors**

Manuscript ID #: NCOMMS-20-20942A

We are grateful to the Reviewers for their constructive comments and believe our paper continues to improve with their valuable input. We agree with all of the Reviewers' suggestions and have responded fully to each point in the revised manuscript (changes are highlighted in blue). Our revisions are described below in a point-by-point manner.

Response Reviewer #1:

I thank Bahmani et al. for responding to the issues I raised on the initial submission. The following substantial concerns remain:

We deeply appreciate the Reviewer continuing to provide valuable feedback on our manuscript. We have strived to address all of the comments as outlined below.

a- Platelet membrane preparation and characterization: referring to a previous publication is not sufficient in my opinion to address the concerns I have been raising regarding the potential presence of EVs. In addition, it should not be taken for granted that the EVs would be, like platelets, emptied by a freeze-thaw cycle. Actually, freezing is one standard way used to store isolated EVs and studies have shown that the vesicular structure, size and count of EV remain unaffected after freeze-thaw. In addition, even if the EVs are indeed disrupted by the freeze-thaw cycle, assuming that their membranes would readily assemble around the nanoparticles in much the same way as platelet membranes is very speculative.

We acknowledge the Reviewer's concerns regarding platelet EVs, which may cause experimental interference. For example, it is possible that the presence of free EVs may serve as inhibitors of PNP binding. We are, however, confident that these concerns can be mitigated based on several factors. First, our initial platelet collection is completed by centrifugation at 2,000 g, a speed that is unlikely to pellet any EVs already present in the PRP. Subsequent membrane collection after the freeze-thaw process is performed at 21,100 g, which based on our experience working with cell membranes will only pellet larger membrane ghosts and fragments. While we cannot guarantee the complete absence of EVs in the membrane preparation, it is also important to note that the coating process involves a sonication step. Even if the freeze-thaw process does not disrupt the EVs, the energy input required for cell membrane coating will rupture any EVs that are present. Finally, based on new TEM imaging provided in **Fig. S1a**, we can see that there is little to no presence of vesicular structures (coated nanoparticles show a clear core-shell structure), which we believe indicates that either (1) there are no EVs present in the final formulation, or (2) EVs are also coated onto the nanoparticle cores. We have included more representative TEM images of our formulation (please refer to our response to the Reviewer's Comment L below) in the revised supplement (see **Page S5**) and further clarified the expected role of sonication for lysing any intact platelet vesicles in the revised text (see **Page 16**)

b- Drug loading efficacy: I thank the authors for providing more detailed experimental information. However, the authors claim a 3.4wt% of R848 encapsulated into the final formulation. How did the

authors verify that this amount is truly encapsulated within the NP and not a residual amount co-purified with the nanoformulation?

We thank the Reviewer for raising this important question. To address this point, in addition to the method to directly quantify drug in the formulation as described in the manuscript, we have now also employed an indirect method to supplement the drug loading study. Specifically, this involved centrifuging the nanoparticles in Amicon filters with a 100 kDa MWCO. Using this method, free unencapsulated drug would pass through the filter and into the filtrate, while encapsulated drug would remain in the retentate with the nanoparticles. The filtrate was analyzed by HPLC and was found to have no detectable levels of free drug.

c- I thank the reviewers for performing the additional experiment of drug release in the absence of 0.05% Triton X-100. The fact that the kinetics of release of R848 from the PMP-R848 is very similar with and without Triton X-100 now questions (a) the need for its addition in the first place and (b) whether the PMP-R848 are truly coated, or maybe imperfectly coated, with platelet membranes... which is worrisome

Originally, Triton X-100 was included in the release kinetics assay in order to prevent any potential interaction between released drug and the container closure. This served to ensure that all released drug could be quantified accurately. After concerns were raised during the first round of review, we later confirmed that the R848 did not, in fact, adhere to the dissolution vessel. Thus, we did not include Triton X-100 in our later studies and instead used 1× PBS at pH 7.4 without additives as our dissolution media. To address the Reviewer's second concern, we have included updated TEM images (please also refer to our response to the Reviewer's Comment L below) of PNP-R848 confirming successful coating of the membrane around the drug-loaded cores. From the images, a distinct core-shell structure can be seen for each nanoparticle, with the membrane layer covering the entirety of the core. This data has been included as **Fig. S1a** in the revised supplement (see **Page S5**).

d- Dialysis method: the authors have answered my concerns in a clear manner

Thank you!

e- I thank the reviewers for documenting that the sterilization of the NPs was achieved by 0.2-micrometre filtration. Was there any loss of NPs at this stage and was it validated that this step does not "uncoat" the PMP-R848?

We appreciate this comment from the Reviewer. As with any purification process, we do expect there to be some loss of nanoparticles during filtration. The PNP-R848 formulation possessed a D95 of ~170 nm after fabrication (i.e. 95% of the nanoparticles were under ~170 nm), as measured by DLS. With 0.2- μ m filtration, some of these particles might be removed. It is important to note that all studies in the manuscript were performed using PNP-R848 post-filtration, and the fact that we were able to successfully confirm the function of the membrane coating should largely alleviate any concerns raised by the Reviewer. Moreover, the updated TEM images in **Fig. S1a** (please refer to our response to the Reviewer's Comment L below) were taken of the final formulation after filtration, and they clearly indicate that the nanoparticles are still fully coated. Through previous studies, it has been theorized that the coating of the

membrane around nanoparticle cores is an energetically favorable process due to the ability of the membrane-associated glycans to promote colloidal stabilization (*Nanoscale* 2014, 6, 2730), which suggests that “uncoating” the nanoparticles would require significant energy input that is unlikely to be supplied by the relatively gentle filtration process. We have clarified this point in the revised text (see **Page 5**) and included a new reference (**ref 45**).

f- Use of human platelets: I understand the argument provided by the authors that the selection of human platelet membrane is logical having in mind considerations for clinical translation. I am also aware that regulatory agencies prefer that pre-clinical studies done within the scope of an IND or registration for a human-tissue product be carried out with the human product itself, especially to assess safety aspects. Such pre-clinical studies are carried out having in mind the risks of immunological reactions from the animal used for such studies. I am doubtful, though, that regulators would be much convinced by the studies carried out in this manuscript... Still, I command the additional in vitro experiments done by the authors but I remain sceptical that they could reflect what happens in vivo. In addition in Fig S2a & b, why are the peaks obtained in flow-cytometry, especially the blank, so broad? Does it reflect a baseline “upregulation”? In Fig S2C, d, & e, showing also mPNP-848 as control would be justified.

We thank the Reviewer for understanding our choice of a human membrane source for our formulation development. We also agree that the results of the in vitro studies, no matter how carefully and methodically designed, cannot be truly reflective of what happens in vivo. In this case, we do believe that the data supports the notion that the immunogenicity of our formulation is driven predominantly by the inclusion of R848. We have attempted to further emphasize this point in the revised discussion (see **Page 8**). Regarding the peaks for the flow cytometry data in **Fig. S2a,b**, we agree that they were indeed broad. Since primary immune cells are sensitive, we believe the higher baseline expression was likely due to the culture and handling conditions during that specific experiment. To address this, we repeated the experiment with extra care to ensure that baseline cell activation was low, and the data in **Fig. S2a,b** has been updated (see **Page S6**). As can be seen, the overall conclusion that the empty nanoformulations do not induce dendritic cell maturation still holds true.

As a side note, we have conducted our pre-IND meeting with the U.S. FDA in September 2020 (**PIND #152044**) to discuss the nonclinical studies and clinical trial plans of our PNP-R848 drug product. We presented our in vivo pharmacology data in the briefing book to the agency and the feedback from the FDA was that “the planned studies appear reasonable”. From a drug development point of view, the regulatory agency is indeed interested in the drug product itself (human platelet in this case) rather than any surrogate drug product (mouse platelet in this case). We sincerely hope the Reviewer can appreciate that, as industrial researchers, our ultimate goal is to advance the PNP-R848 formulation to human clinical trials as a novel drug product.

Figure S2a,b, Expression of CD80 (a) and CD86 (b) by BMDCs after incubation with empty PNP fabricated with mouse platelet membrane (mPNP) or human platelet membrane (hPNP).

g- Statistical analyses: the authors have addressed my concerns.

Thank you!

h- Phosphatidylserine expression and pro-coagulation risks: the authors have addressed my concerns.

Thank you!

i- Dose of 30 uL into 30 - 40 mm³ tumours: the authors provided some words of caution (but it remains a large volume for this tumour size).

Thank you!

j- Tumor volume < 200 mm³: the authors should provide a reference with their answer, or indicate that the value was selected based on their own experimental experience.

The threshold of 200 mm³ was selected based on our own experience with this tumor model. From several repetitions of our experiments, it was observed that mice whose tumors grew larger than this threshold did not survive. This has been clarified in the revised methods section (see **Page 21**).

k- Fig 1. G: early kinetics of release has been added

Thank you!

l- Fig 1.F: The authors provide additional TEM observation. It is quite unclear to my eyes what are the morphological differences between the uncoated and platelet membrane coated nanoparticles. The authors should precisely indicate in figure g where one can observe the platelet membrane. In Fig S1a I am a bit puzzled by the variation in size of the PNP-R848 populations shown. What is actually the variation in size seen in the uncoated starting NPs? Could the apparent heterogeneity in the PNP-R848 population due to the presence of platelet EVs, as discussed above?

We thank the Reviewer for bringing up this important point. Through our extensive experience in working with these cell membrane-coated nanoparticles, we have found that the most indicative sign of a membrane coating is a distinct core-shell structure. A dark ring that forms as a result of the negative stain permeating the hydration layer between the core and membrane can be visualized for coated nanoparticles, whereas it cannot be seen for uncoated cores. We apologize for the confusion regarding the images, which were not of our normal quality due to difficulties in accessing the TEM facility and coordinating sample preparation as a result of the COVID-19 pandemic. In the revised manuscript, we have conducted new TEM imaging and have updated **Fig. 1g**, which now clearly shows the membrane coating compared with the bare cores in **Fig. 1f**. While there is a distribution of sizes among the nanoparticles, as can be seen from the new images in **Fig. S1a**, this is to be expected for polymeric nanoparticles fabricated by self-assembly (*Chem. Soc. Rev.* 2017, 46, 4119; *Chem. Soc. Rev.* 2000, 295; *Front. Bioeng. Biotechnol.* 2019, 7, 259). Note that free vesicles that are not coated around nanoparticle cores have a distinct staining pattern, which we do not see in these images. For the Reviewer's reference, we have included the DLS size distribution profile (see below). The new data and corresponding discussion regarding the visualization of the membrane coating have been included in the revised text (see **Pages 6, 32, and S5**).

Response Fig. 1 Representative DLS size distribution curve for PNP-R848.

Fig. 1f,g, Transmission electron microscopy visualization of uncoated NP-R848 (f) and coated PNP-R848 (g) with uranyl acetate negative staining (scale bars = 50 nm). ***Note the distinct dark ring around the circumference of the nanoparticle cores for (g).

Figure S1a. Transmission electron microscopy visualization (multiple fields of view) of PNP-R848 with uranyl acetate negative staining (scale bars = 100 nm).

m- Fig 1.D: Increase in size: the response from the authors is accepted but remains speculative.

Thank you!

Response to Reviewer #2:

(1) Happy with author(s) change to manuscript title.

Thank you!

(2) Happy with author(s) new (murine and human) cell line panel incl. text and Figure 2.

Thank you!

(3) Not sure I am convinced by data presented in Figure S4 since there appears to be an antitumor effect with PNP without R848 i.e. 25% of mice have tumor < 200 mm³ at 30-days; whilst 100% of the two control groups tumors (Control; PEG-NP) have grown > 200 mm³ at 30-days.

We thank the Reviewer for raising concerns over the antitumor effect of PNP without R848. While there was one mouse with tumor size below endpoint in the PNP-treated group at 30 days, we would like to emphasize that statistical analysis of the Kaplan-Meier survival curves using the log-rank test showed no difference between all groups. We understand that an initial visual inspection of the curves might lead one to reach a different conclusion, however, it should also be acknowledged that in vivo experiments carry inherently large variability. We acknowledge that, based on this experiment, we cannot rule out with complete certainty that there is no effect from empty PNP treatment. However, we are confident that, if there was indeed a difference, the effect size for the PNP group versus the control would be small. Ultimately, we believe that the more important point to focus on is that, when loaded with R848, the PNP-R848 formulation unequivocally produces a large and statistically significant effect that can be easily discerned, as can be seen **Fig. 4d and 6d**. To address this comment, in the revised text, we have attempted to further clarify the situation in the discussion for **Fig. S5** (see **Page 10**).

(4) Happy with author(s) new antitumor efficacy data in 4T1 model highlighted in text and Figure 6.

Thank you!

(5) New PNP-R848 PK data (Fig. 2e,f,) is welcome. However, it is hard to interpret the relevance of the result since no information is provided on the PNP-R848 dose used in Fig. 2e,f,. Moreover, how this dose relates to the PNP-R848 dose used in the MC38 efficacy study (Fig. 4.). This dataset would be further improved with respect to safety risk if systemic cytokines had been included.

We apologize for not specifically providing the dosage information for the PK data. We would like to clarify that for the PK study we used the same dosage (15 μg of R848 per injection) as the efficacy study. This information has been updated in the revised methods section (see **Page 21**).

With regards to the systemic cytokine burden after administration, we conducted an additional experiment to study this parameter after intratumoral administration of PNP-R848. From the data, it can be seen that there was a transient elevation of pro-inflammatory markers such as IL-6, TNF- α , and IL-12. Importantly, the serum levels of all the cytokines that were studied quickly returned back to baseline levels. Observationally, the mice did not exhibit any adverse effects within this 48-hour period, and the body weight data in **Fig. 4e** also suggests that there was no major toxicity, even after repeated dosing. In the clinic, a transient elevation of cytokine levels is generally considered normal for patients receiving immunotherapies (*Blood* 2014, 124, 188; *Annu. Rev. Med.* 2020, 71, 47). We should clarify that the safety benefit of our approach derives mostly from our use of intratumoral administration, which allows us to significantly decrease the drug dosage compared with systemic therapies (*Clin. Cancer Res.* 2014, 20, 1747). At the same time, the enhanced retention of PNP-R848 at the tumor site drives improved antitumor efficacy compared with untargeted or free-drug formulations. To address the Reviewer's concern, we have included the serum cytokine data as **Fig. S4** and updated the relevant discussion and methods (see **Pages 10, S3, and S8**). We have also modified the language in the introduction and discussion to clarify that systemic leakage of free drug after intratumoral injection is likely to yield reduced efficacy (see **Pages 4 and 7**).

Figure S4. Systemic cytokine levels. a-c, Levels of IL-6 (a), TNF α (b), and IL-12p40 (c) in the serum of mice after intratumoral treatment with PNP-R848 (mean \pm SD).

(6) Happy with author(s) additional TLR8 reporter data incl. text Fig. 3.

Thank you!

(7) Happy with author(s) additional cytokine data incl. text Fig. 3.

Thank you!

Response to Reviewer #3

The revised manuscript has addressed all the comments and questions raised by the reviewers and I am satisfied with the answers. Therefore, it's OK for acceptance now.

We are grateful to the Reviewer for taking the time to review our manuscript and provide valuable feedback.

REVIEWERS' COMMENTS

Reviewer #1 (Remarks to the Author):

The authors have made substantial efforts to address the concerns I have raised along the review process although some questions remain to be solved in future developments

**Intratumoral immunotherapy using platelet-cloaked nanoparticles enhances
antitumor immunity in in solid tumors**

Manuscript ID #: NCOMMS-20-20942B

We are grateful to the Reviewers for their constructive comments throughout this process.

Response Reviewer #1:

The authors have made substantial efforts to address the concerns I have raised along the review process although some questions remain to be solved in future developments

We would like to thank the Reviewer for spending their valuable time to carefully go over our revised manuscript. Their support is much appreciated.